# A comprehensive study of deep learning for soil moisture prediction

Yanling Wang[1], Liangsheng Shi[1*], Yaan Hu[2], Xiaolong Hu[1], Wenxiang Song[1], Lijun Wang[1]

[1]State Key Laboratory of Water Resources Engineering and Management, Wuhan University, Wuhan, China

[2]State Key Laboratory of Hydrology-Water Resources and Hydraulic Engineering, Nanjing Hydraulic Research Institute, Nanjing, China

*Correspondence to*: Liangsheng Shi (liangshs@whu.edu.cn)

**Abstract.** Soil moisture plays a crucial role in the hydrological cycle, but accurately predicting soil moisture presents challenges due to the nonlinearity of soil water transport and variability of boundary conditions. Deep learning has emerged as a promising approach for simulating soil moisture dynamics. In this study, we explore ten different network structures to uncover their mechanisms of data utilization and maximize the potential of deep learning for soil moisture prediction, including three basic feature

extractors and seven diverse hybrid structures, six of which are applied to soil moisture prediction for the first time. We compare the predictive abilities and computational costs of the models across different soil textures and depths systematically. Furthermore, we exploit the interpretability of the models to gain insights into their workings and attempt to advance our understanding of deep learning in soil moisture dynamics. For soil moisture forecasting, our results demonstrate that the temporal modeling capability

of Long Short-Term Memory (LSTM) is well-suited. Besides, the improved accuracy achieved by feature attention LSTM (FA-LSTM) and the generative adversarial network-based LSTM (GAN-LSTM), along with the Shapley additive explanations (SHAP) analysis, help us discover the effectiveness of attention mechanisms and the benefits of adversarial training in feature extraction. These findings provide effective network design principles. The Shapley values also reveal varying data leveraging approaches

among different models. The t-Distributed Stochastic Neighbor Embedding (t-SNE) visualization illustrates differences in encoded features across models. In summary, our comprehensive study provides insights into soil moisture prediction and highlights the importance of the appropriate model design for specific soil moisture prediction tasks. We also hope this work serves as a reference for deep learning studies in other hydrology problems. The codes of 3 machine learning and 10 deep learning models are

open sourced.

***Key words***: Soil moisture; deep learning; hybrid models; machine learning; forward modeling; model interpretability

**1.Introduction**

Soil moisture is significant in simulating many hydrological processes since it controls the interaction of water and energy between the land surface and the atmosphere (Entin et al., 2000; Vereecken et al., 2022). Accurately providing information on soil moisture dynamics is crucial for effective water resources planning and management, agricultural production, climate prediction, and flood disaster monitoring (Vereecken et al., 2008; Sampathkumar et al., 2013). However, caused by the randomness of

rainfall and the nonlinear features of infiltration and evaporation processes (Guswa et al., 2002), soil moisture is highly variable and nonlinear in space and time (Heathman et al., 2012), which makes it difficult to forecast.

    Since various mainstream approaches have been applied for soil moisture dynamics prediction, a comprehensive study is needed to provide suitable solutions for different predicting tasks, encourage

improvements on models and build confidence in this area. Traditionally, soil moisture dynamics prediction is widely based on physical models, such as the soil-plant-air model (Saxton et al., 1974), HYDRUS (Simunek et al., 2005), and CATHY (Camporese et al., 2015). Though these models are interpretable, they perform poorly in practical applications, because of the inestimable parameters (Gill et al., 2006) and inadequate description of physical processes (Li et al., 2022b). With the reduction in

data acquisition costs and advancements in computation, there has been an increasing focus on data-driven models. Initially, multiple linear regression (Qiu et al., 2003; Hummel et al., 2001) and empirical models (Azhar et al., 2011; Verma and Nema, 2021) are applied for soil moisture prediction. However, one nonnegligible problem is that these methods require calibrations and have limited generalization capabilities (Holzman et al., 2017; Jackson, 2003). Compared to these traditional data-driven models,

machine learning methods appear to possess stronger data fitting ability. For instance, support vector regression (SVR) (Gill et al., 2006) and random forest (RF) (Prasad et al., 2019) have both shown satisfactory and robust results with low computing costs in soil moisture prediction. Additionally, the single-layer feedforward neural network with generalized inverse operation -- Extreme Learning Machine (ELM) (Huang et al., 2006) can precisely predict the future trends of soil moisture and support

future irrigation scheduling. (Liu et al., 2014). What's more, when dealing with multi-scale soil moisture

data, such as satellite data, Abbaszadeh et al. employed 12 distinct Random Forest models to downscale the daily composite version of SMAP data (Abbaszadeh et al., 2019).

Currently, deep learning is the state-of-the-art data-driven method, which has made obvious improvements in many research areas (Lecun et al., 2015). Due to their powerful approximation ability, deep neural networks (DNNs) (Goodfellow et al., 2016) have been extensively researched from soil moisture descriptions (Cai et al., 2019; Prakash et al., 2018). Notably, recurrent neural networks (RNNs) (Pollack, 1990) excel at capturing temporal information in time series data and model sequential dependencies for predictions (Mikolov et al., 2011). This is consistent with the characteristics of soil moisture dynamics simulation. Fang et al. (2019) utilized Long Short-term Memory (LSTM) (Hochreiter and Schmidhuber, 1997) for soil moisture and received satisfactory results. Besides, Sungmin O et al. efficiently employed LSTM to interpolate global gridded datasets from in-situ observations (Orth, 2021; Orth et al., 2022). From a different perspective, Convolution Neural Networks (CNNs) (LeCun, 1989) are capable of extracting features from training data in specific dimensions, making them widely used in dealing with 2-D (Albawi et al., 2018; Patil and Rane, 2021) or 1-D data (Severyn and Moschitti, 2015; Shi et al., 2015). Therefore, 1D-CNNs are applied in many hydrology researches(Hussain et al., 2020; Chen et al., 2021). Additionally, attention mechanisms enable the selection of critical information from multiple input features or model outputs, which can be visualized using attention weight (Ding et al., 2020; Li et al., 2022a). On this foundation, self-attention can model dependencies and aggregate features from inputs disregarding their distance (Vaswani et al., 2017), which shows great potential in soil moisture prediction.

As various deep learning approaches focused on distinct mechanisms of data utilization, hybrid structures become a vital research area. On one hand, combining the feature importance processing methods -- attention mechanisms, with deep learning models, can indeed lead to improvements (Ahmed et al., 2021; Ding et al., 2019; Kilinc and Yurtsever, 2022). Li et al. proposed an attention-aware LSTM to estimate soil moisture and temperature and achieved better performance than LSTM alone (2022). In their work, three attention mechanisms help obtain the spatial-temporal feature vectors of LSTM inputs or outputs. On the other hand, the combinations of multiple neural networks tend to perform better than a single network alone (Semwal et al., 2021). The hybrid CNN-GRU model proposed by Yu et al.(2021) outperformed the independent CNN or GRU model in predicting root zone moisture. Besides, Li, et al. (2022) proposed EDT-LSTM, a stacked LSTM model based on the encoder-decoder structure (Sutskever

et al., 2014) and residual learning (He et al., 2016). This achieved more stable results than a single LSTM. Regarding the optimization of training strategies, adversarial training in generative adversarial networks (GANs) (Goodfellow et al., 2014) can capture more information on real data. This helps to address the problem of fuzzy prediction and provides a superior solution for weather forecasts (Jing et al., 2019; Ravuri et al., 2021). Moreover, advancements in model structure have been instrumental in enhancing performance and improving generalization abilities. For instance, Liu et al integrated multi-scale designs into their models (Liu et al., 2022). In addition to pure deep learning models, differentiable, physics-informed machine learning models with a physical foundation have emerged as a noteworthy development. This kind of model systematically integrates physical equations with deep learning, enabling the prediction of untrained variables and processes with high accuracy (Feng et al., 2023).

Therefore, it is essential to design effective and suitable neural network structures for soil moisture prediction tasks. In this study, we comprehensively evaluate the performance of various deep learning methods in soil moisture prediction, highlighting their key characteristics in terms of prediction accuracy and computational costs. The models evaluated in this research range from machine learning models such as RF, ELM, and SVR to basic deep learning models, including 1D-CNN, LSTM, and the encoder of Transformer(Vaswani et al., 2017), and hybrid deep learning models, including CNN-LSTM, LSTM-CNN, CNN-with-LSTM, FA-LSTM, TA-LSTM, FTA-LSTM, and GAN- LSTM. Notably, the encoder of the Transformer is first developed in soil moisture prediction, with CNN-LSTM, LSTM-CNN, FA-LSTM, TA-LSTM, FTA-LSTM, and GAN-LSTM first applied and systematically compared for soil moisture. To gain insights into their workings and provide a thorough analysis of why some methods perform better, we utilize the SHAP (Lundberg et al., 2018) method to demonstrate the importance of features in different models and employ t-SNE visualization (Van der Maaten and Hinton, 2008) to show the encoded features across models. The systematical assessment of the models is carried out across multiple sites at 5 depths. For forecasting soil moisture, the utilized data include meteorological data, soil temperature data, and soil moisture content data from previous days, as these inputs are closely associated with evaporation and infiltration processes.

In the remainder of this article, Sect. 2 describes the data used and the deep learning background; Sect. 3 presents a detailed description of the participating methods; Sect. 4 analyzes comparison results and discusses the interpretability of the models. The conclusion is drawn in Sect. 5.

## 2. Data Description and Backgrounds

### 2.1 Data Description

To create a comprehensive evaluation under different soil types, in-situ observations at thirty different sites are downloaded from the International Soil Moisture Network (ISMN) (https://ismn.geo.tuwien.ac.at/en/). The research sites are carefully chosen according to the geographical location (dispersed as much as possible), soil textures, and distinct land cover types (diverse as much as possible). The spatial locations of the sites are shown on a world map in Fig. 1. More detailed site meteorological information and soil moisture time series data are provided in Appendix D.

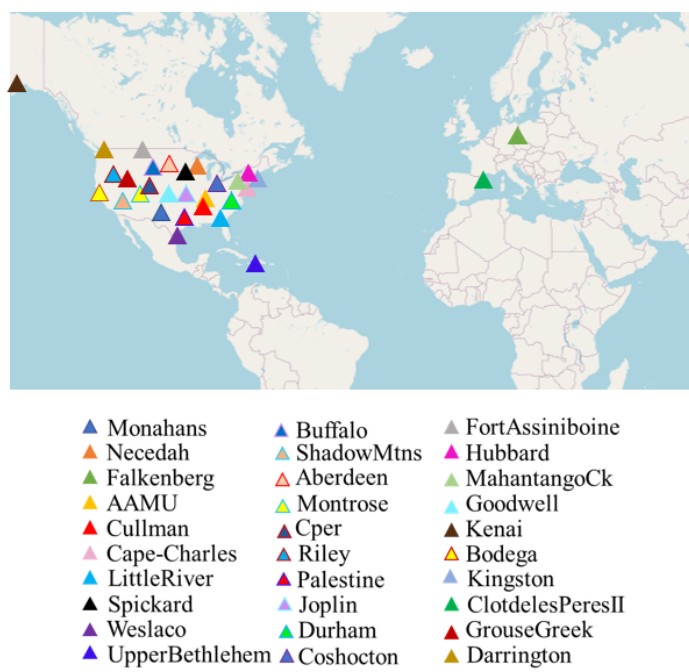

**Figure 1.** The spatial locations and soil moisture content time series at various depths of thirty sites.

In the process of input factor screening, we carefully choose meteorological inputs based on the precipitation and evapotranspiration calculation, including precipitation (P), atmospheric temperature(AT), long-wave radiation(LR), short-wave radiation(SR), wind speed(WS), and relative humidity(RH)), which are closely related to the soil evapotranspiration and infiltration processes. Besides, soil temperature(ST) data, along with soil moisture data from the previous day(SM) are incorporated to represent the soil condition. Fig. 2 displays the Pearson correlation analysis results for input factors at the Cape-Charles and UpperBethlem sites. Pearson correlation analysis examines the relationship between two variables by calculating the correlation coefficient, measuring the strength and direction of

their association. Notably, the correlation coefficients between soil moisture and the input data vary greatly with both the station and depth. While the correlation coefficient between longwave radiation and soil moisture is low at UpperBethlem site, it is significant at Cape-Charles, highlighting the influence of site-specific differences. Although utilizing highly correlated factors as inputs appears to be a logical choice, achieving uniformity across different sites and depths can be difficult. This presents a crucial aspect to explore when evaluating the performance of models for self-learning screening of significant influencing factors. Therefore, the input data $x_t$ at time $t$ consists of all eight factors, $x_t = \{P_t, T_t, LW_t, SW_t, RH_t, WS_t, ST_t, SM_{t-1}\}$. Since groundwater level observations are difficult to obtain, changes in the lower boundary conditions are excluded from the inputs.

Fig. 3 shows the autocorrelation analysis conducted at 5 soil depths. The autocorrelation coefficients for soil water content at different depths decrease with increasing delay days. The most significant change is observed in the surface layer. As a result, we have utilized a 4-day delay as our input for all deep learning models in this study to forecast the soil moisture content on the fifth day. This means the input vector $I, \{x_{t-3}, x_{t-2}, x_{t-1}, x_t\}$ is used to predict the target value $y_t$, that is the soil moisture $SM_t$ at time $t$. For machine learning, we only utilize the $x_t$ to generate predictions.

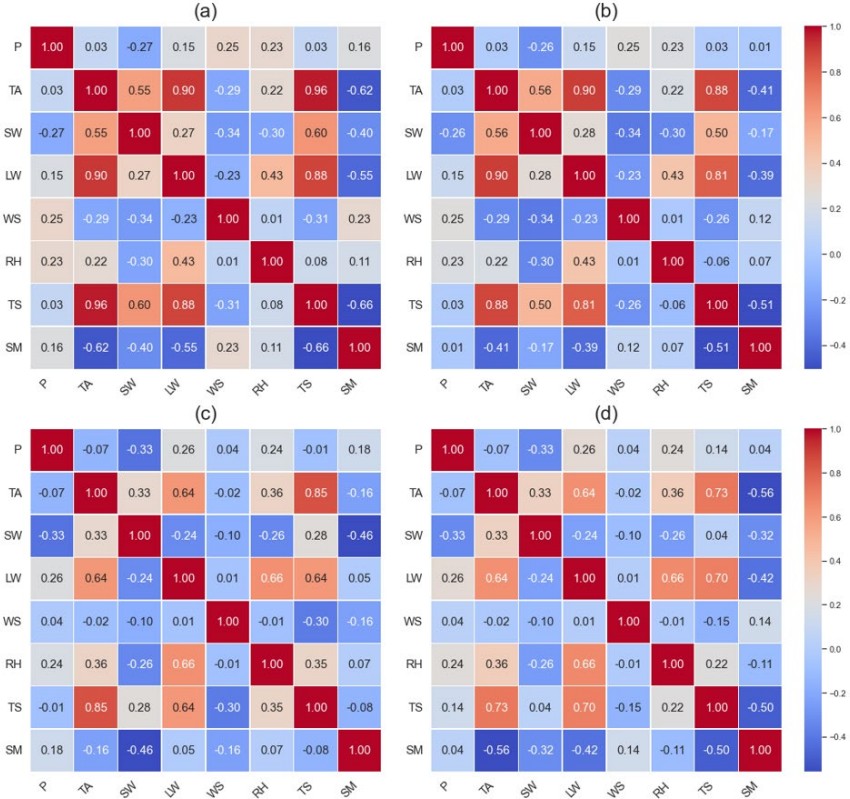

**Figure 2**. Pearson correlation analysis results among the observed variables of 0.05m and 1.00m at Cape-Charles (a) (b) and UpperBethlem (c) (d) sites.

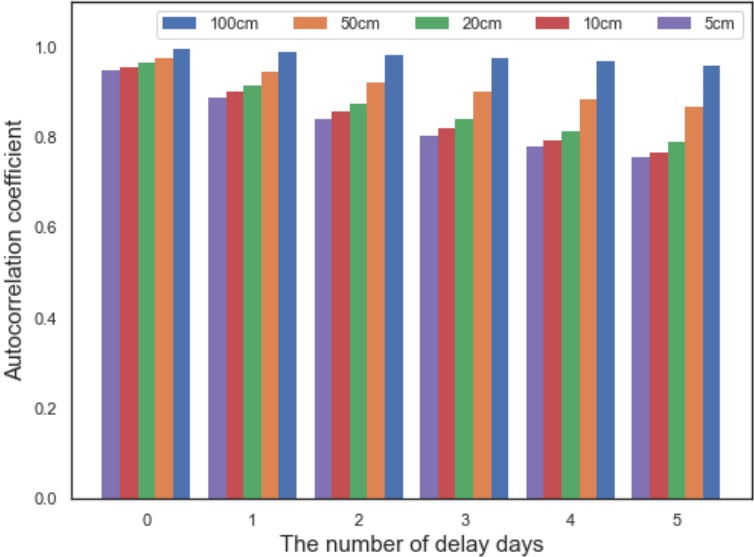

**Figure 3**. Autocorrelation analysis results of soil water content with different days delay at Cape-Charles.

This study builds individual predictive models for each site and depth, disregarding the inclusion of static properties such as land cover, soil hydraulic properties, and topography. Soil moisture and soil

temperature data are obtained from the ISMN. Specifically, the meteorological data applied in this work is sourced from the NASA POWER project (https://power.larc.nasa.gov/), which provides a wide range of meteorological data, including temperature, precipitation, solar radiation, and more. Detailed information can be found at (https://power.larc.nasa.gov/docs/methodology/data/sources/). The meteorological data are used as an auxiliary component for soil moisture prediction in our work.

Therefore, even though the resolution of some variables appears coarse, we can safely disregard the potential influence of resolution on our research findings and conclusions.

**Table 1.** Summary of main characteristics of thirty sites.

| | Sand | Silt | Clay | Land cover | Period | Lat. | Lon. |
|---|---|---|---|---|---|---|---|
| Kingston-1-W | 85 | 10 | 5 | Grassland | 2012-2023 | 41.48 | -71.54 |
| HubbardBrook | 85 | 11 | 4 | Tree cover | 2003-2022 | 43.93 | -71.72 |
| Monahans-6-ENE | 83 | 6 | 11 | Shrub cover | 2010-2022 | 31.62 | 102.81 |
| Necedah-5-WNW | 83 | 11 | 6 | Grassland | 2009-2022 | 44.06 | -90.17 |
| Shadow Mtns | 79 | 10 | 11 | Shrub cover | 2013-2017 | 35.47 | -115.72 |
| Falkenberg | 73 | 21 | 6 | Cropland, rained | 2003-2020 | 52.17 | 14.12 |
| Kenai-29-ENE | 54 | 38 | 8 | Shrub cover | 2012-2023 | 60.72 | -150.45 |
| AAMU-jtg | 53 | 22 | 25 | Grassland | 2010-2022 | 34.78 | -86.55 |
| Darrington-21-NNE | 53 | 22 | 25 | Tree cover | 2013-2019 | 48.54 | -121.45 |
| Palestine-6-WNW | 49 | 27 | 24 | Grassland | 2009-2013 | 31.78 | -95.72 |
| Durham-11-W | 49 | 27 | 24 | Herbaceous cover | 2009-2016 | 40.37 | -81.78 |
| Cullman | 49 | 27 | 24 | Mosaic Cropland | 2006-2022 | 34.20 | -86.80 |
| Cape-Charles | 49 | 27 | 24 | Herbaceous cover | 2011-2022 | 37.29 | -75.93 |
| LittleRiver | 47 | 30 | 23 | Grassland | 2005-2020 | 31.50 | -83.55 |
| Montrose-11-ENE | 43 | 35 | 22 | Tree cover | 2010-2023 | 38.54 | -107.69 |
| Coshocton-8-NNE | 41 | 39 | 20 | Grassland | 2009-2016 | 40.37 | -81.78 |
| MahantangoCk | 41 | 39 | 20 | Cropland | 2002-2021 | 40.67 | -76.67 |
| Bodega-6-WSW | 39 | 38 | 23 | Grassland | 2011-2023 | 38.32 | -123.08 |
| GrouseGreek | 36 | 41 | 23 | Grassland | 2016-2023 | 41.78 | -113.82 |
| Aberdeen-35-WNW | 36 | 41 | 23 | Grassland | 2012-2023 | 45.71 | -99.13 |
| Goodwell-2-SE | 36 | 41 | 23 | Grassland | 2010-2022 | 36.57 | -101.61 |
| FortAssiniboine#1 | 36 | 41 | 23 | Grassland | 2017-2021 | 48.48 | -109.8 |
| Cper | 36 | 41 | 23 | Grassland | 2013-2021 | 40.82 | -104.71 |
| Riley-10-WSW | 36 | 41 | 23 | Shrub cover | 2011-2021 | 43.47 | -119.69 |
| Spickard | 35 | 41 | 24 | Grassland | 2010-2022 | 40.25 | -93.72 |
| Joplin-24-N | 35 | 41 | 24 | Grassland | 2010-2020 | 37.43 | -94.58 |
| Weslaco | 34 | 45 | 21 | Cropland, rained | 2017-2021 | 26.16 | -97.96 |
| UpperBethlehem | 32 | 38 | 30 | Herbaceous cover | 2008-2010 | 17.72 | -64.80 |
| Buffalo-13-ESE | 31 | 44 | 25 | Grassland | 2012-2023 | 45.52 | -103.30 |
| ClotdelesPeresII | 19 | 49 | 32 | Cropland | 2021-2023 | 42.16 | 0.84 |

## 2.2 Deep Learning Backgrounds

Deep learning enhances the complexity and learning capability of traditional machine learning methods by adding multiple layers (Kamilaris and Prenafeta-Boldú, 2018). At each layer, input signals are weighted through the connections of each neuron and subsequently activated by activation functions (Schmidhuber, 2015). Deep learning discovers intricate structures in training data by utilizing backpropagation to guide the machine in adjusting its internal parameters (Lecun et al., 2015).

In this study, the primary challenge in soil moisture prediction is processing the time-series data with specific dimensions and simulating soil moisture dynamics with high spatiotemporal variability. Given the diversity of neural networks, numerous methods have the potential to deal with specific time-series data. CNNs can extract local temporal information from the data by sliding convolutional kernels along the time dimension. On the other hand, RNNs excel at capturing the overall temporal sequence information. Additionally, self-attention has the potential to associate inputs and make predictions, making them capable of handling sequence data effectively. These three types of networks can be regarded as fundamental feature extractors in deep learning. Furthermore, hybrid deep learning models integrate the characteristics of multiple models, enhancing their prediction capacities (Yu et al., 2021). Combinations of CNNs, RNNs, and attention mechanisms have been widely utilized in many studies. Besides, employing specified training strategies with suitable network structures can also improve prediction performance. For instance, GANs enable the training objective of neural networks to go beyond minimizing data mean squirt error and utilize adversarial training to fully capture data regularities. By designing appropriate network structures and training strategies, it is possible to further improve prediction accuracy.

It is necessary to conduct a comprehensive evaluation to analyze the internal combining meaning of models and decide the most suitable combination rule for soil moisture prediction. With the collected data in 2.1, it is possible to deeply explore the prediction abilities of the deep learning models. We evaluate models from the perspectives of prediction accuracy and computational costs to provide a reference for soil moisture dynamics predictions. Further research on model interpretability can provide insights into how the model structure influences the utilization of data, leading to a more effective design of the model structure.

**3.Models and Methodology**

Three machine learning models and seven deep learning models take part in this comparative research. Introductions to each model are provided below, along with key references for interested readers. The parameters of each model are recorded in Appendix A.

**3.1 Machine Learning Methods**

In this study, machine learning models Random Forest (RF), Extreme Learning Machine (ELM), and Support Vector Machine (SVM) are applied to compare with the deep learning models as a benchmark.

Random Forest, proposed by Breiman (Breiman, 2001), is used for regression and classification tasks and has gained popularity for its high accuracy. RF works by constructing multiple decision trees on randomly sampled subsets of the training data. Each tree is trained on a random subset of features, and the final prediction is made by averaging the predictions of the individual trees. This approach reduces overfitting and increases model stability. For soil moisture prediction, RF has proven to be a stable and reliable method (Carranza et al., 2021).

Extreme Learning Machine (Huang et al., 2006) utilizes a single-layer feedforward neural network as its foundation. ELM achieves fast learning speed and strong generalization ability by employing random input layer weights and biases and applying generalized inverse matrix theory to calculate the output layer weights. The algorithm has been applied in various fields and has shown promising results. Liu et al.(2014) employed ELM to predict the large-scale soil moisture in Australian orchards. The results demonstrated that the model was capable of accurate forecasting.

Support Vector Machine (Cortes and Vapnik, 1995) was proposed for applications in classification and regression. It aims to find the maximum-margin hyperplane that best separates sample points. To make this hyperplane more robust in high-dimensional feature spaces, SVM uses kernel functions to perform nonlinear mapping and create a new feature space where the data can be linearly separable. The algorithm then finds the optimal classification hyperplane with the maximum margin. SVMs have achieved great success in various fields. Gill et al. (2006) applied SVM to soil moisture prediction and compared it with DNNs. The results showed that SVM was suitable for soil moisture content prediction. Support Vector Regression (SVR) is a variant of SVM that is specifically designed for regression tasks, which is applied in this study.

For machine learning, $x_t$ and $y_t$ represents the input feature and target object, respectively. The input data corresponds one-to-one in time to the target and serves as both the input and output of the

machine learning models. The prediction accuracy of machine learning serves as a comparison for deep learning models. Hyperparameters used in models are recorded in Appendix A.

## 3.2 Basic Deep Neural Networks

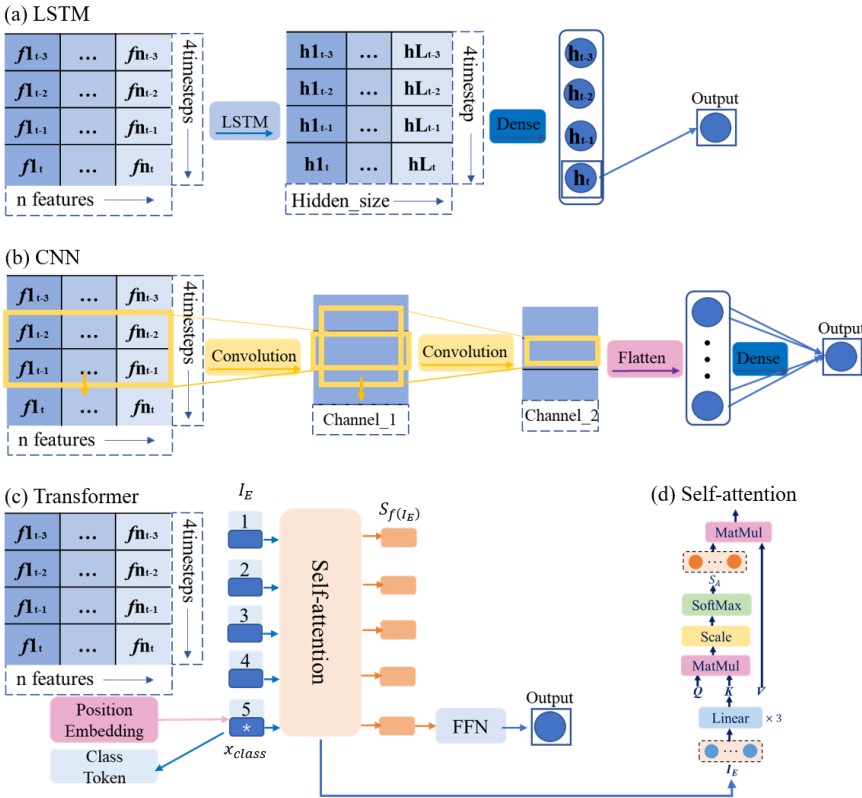

**Figure 4.** Network structures of the LSTM(a), the 1D-CNN(b), and the proposed Transformer (c) inspired by Dosovitskiy et al.(2020) with the self-attention structure (d)for soil moisture prediction.

### 3.2.1 LSTM

RNNs (Pollack, 1990) operate by recursing in the direction of sequence progression, with all nodes in the network being chained together. These unique properties make RNNs effective in processing sequence data and extracting temporal information, which has led to breakthroughs in natural language processing (Connor et al., 1994). The ability of RNNs to model temporal dependencies is suitable for predicting soil moisture.

Long Short-Term Memory (LSTM) (Hochreiter and Schmidhuber, 1997) neural networks, were proposed to address the limitations of traditional RNNs. LSTM can overcome the issue of gradient

vanishing and memorize more useful information through a special unit, which is called the cell state. Thus, LSTM operates as follows:

$$i_t = \sigma(W_i \cdot [h_{t-1}, x_t] + b_i) \tag{1}$$

$$f_t = \sigma(W_f \cdot [h_{t-1}, x_t] + b_f) \tag{2}$$

$$o_t = \sigma(W_o \cdot [h_{t-1}, x_t] + b_o) \tag{3}$$

$$\tilde{C}_t = tanh(W_c \cdot [h_{t-1}, x_t] + b_c) \tag{4}$$

$$c_t = f_t \cdot c_{t-1} + i_t \cdot \tilde{C}_t \tag{5}$$

$$h_t = o_t \cdot tanh(c_t) \tag{6}$$

where $W_i$ and $b_i$ are the parameters for the input gate, $W_f$ and $b_f$ are the parameters for the forget gate, $W_o$ and $b_o$ are the parameters for the output gate, $W_c$ and $b_c$ are used for cell state updating; $\sigma$ is the activation function.

We generate the time-dependent hidden states $\boldsymbol{H}$, $\{\boldsymbol{h}_{t-3}, \boldsymbol{h}_{t-2}, \boldsymbol{h}_{t-1}, \boldsymbol{h}_t\}$ from input $\boldsymbol{I}, \{\boldsymbol{x}_{t-3}, \boldsymbol{x}_{t-2}, \boldsymbol{x}_{t-1}, \boldsymbol{x}_t\}$ through the LSTM. After sequentially processing all inputs in the LSTM, the last hidden state $\boldsymbol{h}_t$ of the sequential output is used as the prediction for network training, as depicted in Fig. 4a. This is because the input features at each time step can be encoded in the last hidden state. The parameters in this model are recorded in Appendix A.

### 3.2.2 1D-CNN

CNNs (LeCun, 1989) were originally applied for image recognition. The convolution and pooling layers in CNNs can extract the distinguishing features of the given data while reducing the amount of data to be processed (Ajit et al., 2020). Consequently, CNNs are highly effective in processing data that come in the form of multiple arrays.

For time series data, 1D-CNNs can extract local temporal features via convolution kernels that slide along the time dimension. 1D-CNNs have demonstrated success in speech and natural language processing applications (Abdel-Hamid et al., 2014; Severyn and Moschitti, 2015). Hence, 1D-CNNs are capable of soil moisture prediction tasks. The complete forward-propagation process of a simple 1D-CNN for soil moisture prediction is illustrated in Fig. 4b. Given that the input vector $I, \{\boldsymbol{x}_{t-3}, \boldsymbol{x}_{t-2}, \boldsymbol{x}_{t-1}, \boldsymbol{x}_t\}$, two convolution layers are employed in the 1D-CNN architecture. The

convolution kernel size (Kernel_size) is set to 2, with a stride of 1. Specific parameters are listed in Table

A1. To preserve the information of the data, pooling layers are intentionally omitted.

### 3.2.3 Transformer

The self-attention mechanism can model the dependencies and aggregate features from inputs. Therefore, a stacking structure of self-attention mechanisms like Transformer (Vaswani et al., 2017)

can achieve the functions of CNNs and RNNs without iterations. This provides a novel way for predictions. In this study, we utilize the encoder structure of the Transformer (Vaswani et al., 2017), as depicted in Fig. 4c, to predict soil moisture. The self-attention is shown in Fig. 4d, which operates as follows:

$$S_A = softmax\left(\frac{QK^T}{\sqrt{d_k}}\right) = softmax\left(\frac{(W_Q I_E)(W_K I_E)^T}{\sqrt{d_k}}\right) \tag{7}$$

$$S_{f(I_E)} = S_A \otimes V = softmax\left(\frac{(W_Q I_E)(W_K I_E)^T}{\sqrt{d_k}}\right) W_V I_E \tag{8}$$

where $W_K$, $W_V$ and $W_Q$ are the key, value, and query parameter matrices, respectively; $I_E$ is the

Transformer input; $\frac{1}{\sqrt{d_k}}$ is the scaling factor, $d_k = 4$.

The outputs generated by the self-attention mechanism correspond to the inputs one-to-one. In this study, a "class token" vector $x_{class}$ is introduced as additional input to start the prediction process. The class token is randomly initialized and can be trained, serving as the fifth input. It enables aggregate global features from all other inputs and avoids bias towards a specific time step in the sequence.

However, the self-attention mechanism ignores the temporal order of the inputs. To address this issue, we incorporate positional encoding to preprocess the inputs. Both the learnable positional encoding and sine cosine coding are tested in this research. The sine cosine positional encoding is defined as:

$$PE_{(pos1,2pos2)} = sin\left(\frac{pos1}{10000^{\frac{2pos2}{d_{model}}}}\right) \tag{9}$$

$$PE_{(pos1,2pos2+1)} = cos\left(\frac{pos1}{10000^{\frac{2pos2}{d_{model}}}}\right) \tag{10}$$

where the parameters $pos1$ and $pos2$ represent the positions of the first and second dimensions of the input, respectively. Here, $d_{model} = 8$ denotes the parameter of self-attention, which is equal to the input

features at each time step.

The encoded position vectors $PE$ are added to the original inputs before feeding them into the Transformer. With $PE$, the input of the Transformer is defined as follows:

$$I_E = \{\boldsymbol{x}_{t-3}, \boldsymbol{x}_{t-2}, \boldsymbol{x}_{t-1}, \boldsymbol{x}_t, \boldsymbol{x}_{class}\} + PE \qquad (11)$$

### 3.3 Hybrid Deep Learning Models

### 3.3.1 Hybrid structure of CNN and LSTM

In this section, three connecting ways of CNNs and LSTMs, CNN-LSTM, LSTM-CNN, and CNN-with-LSTM, are considered. These hybrid models possess advanced capabilities in handling diverse types of data, generally leading to improved prediction accuracy. To ensure a rigorous comparison with the previous 1D-CNN and LSTM models, the parameters of the CNN and LSTM layers in our hybrid

models are kept as consistent as possible with the 1D-CNN and LSTM models. The detailed parameter setting information can be found in Table A1.

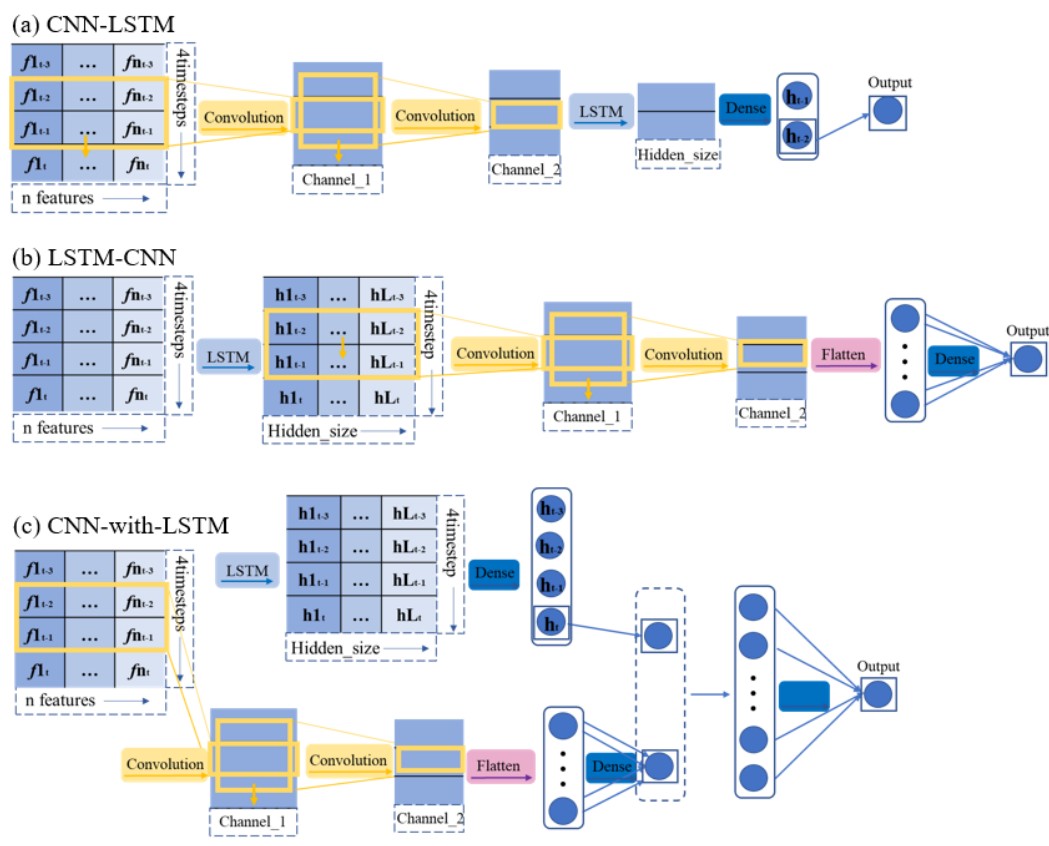

**Figure 5.** The framework of the proposed CNNs and LSTMs hybrid models: CNN-LSTM(a), LSTM-CNN(b), and CNN-with-LSTM(c).


**CNN-LSTM**

Generally, the CNN-LSTM model is comprised of CNN layers followed by LSTM layers. The input data first passes through convolution layers to better extract local features in the sequence data. Then LSTM layers are used to associate the time-series extracted features. Therefore, this kind of model excels at handling the input data in image format, which has been widely utilized in prediction tasks, yielding positive outcomes in various applications (Semwal et al., 2021). In our soil moisture prediction task, CNN-LSTM consists of 2 convolution layers and an LSTM layer, which is shown in Table A1. As we mentioned in Section 3.2, the last hidden state $h_t$ is still applied as the prediction. Fig. 5a depicted the structure of CNN-LSTM applied in this research.

**LSTM-CNN:**

In contrast to the CNN-LSTM model, the LSTM-CNN model first utilizes LSTM layers to associate the time series data and output high-dimensional related hidden states. Subsequently, convolution layers are employed to extract the features of these time-dependent hidden states. This model has also been widely adopted in various applications (Xia et al., 2020). In this study, LSTM-CNN for soil moisture prediction consists of an LSTM layer and 2 convolution layers sequentially. The structure of LSTM-CNN can be seen in Fig. 5b. Detailed layers and parameters of this model are presented in Table A1.

**CNN-with-LSTM:**

CNN-with-LSTM is a model that employs the parallel combination of both CNN and LSTM, merging their outputs through concatenation, and uses a fully connected network for regression analysis. By combining the feature extraction capabilities of CNN with the time series memory ability of LSTM, this model captures both the local and global temporal characteristics of the input data. This kind of hybrid structure has been used in soil moisture prediction and achieved satisfactory results(Yu et al., 2021). In our work, CNN-with-LSTM is comprised of an LSTM layer and 2 convolution layers parallelly, and the structure is depicted in Fig. 5c. Table A1 lists the network structures of the CNN and LSTM models in addition to the parameter settings.

### 3.3.2 Hybrid Structure of Attention and LSTM

To enhance the accuracy of deep learning models and address the issue of lack of interpretability, attention mechanisms have been incorporated into LSTM models to weigh the importance of different

input and output vector dimensions (Li et al., 2022a; Ding et al., 2020; Xia et al., 2020). Attention mechanisms are commonly used in combination with other neural networks as a form of pre-processing or post-processing. Through training, attention mechanisms dynamically generate spatiotemporal attention importance weights to selectively focus on critical parts of the input or output, as illustrated in Fig. 6. These attention weights enable the model to assign importance to various elements within the input sequence, thus helping make more accurate predictions. Additionally, these attention weights offer a visualized representation, which provides insights into the sections of the input sequence most essential for a specific prediction. According to the specific roles of the attention mechanisms, the hybrid models can be classified into three categories: FA-LSTM (a feature attention mechanism with LSTM), TA-LSTM (a temporal attention mechanism with LSTM), and FTA-LSTM (an LSTM combines both feature and temporal attention mechanisms). Ding et al.(2020) conducted experiments on these three kind of hybrid models in flood prediction, confirming the effectiveness of incorporating LSTM with attention mechanisms.

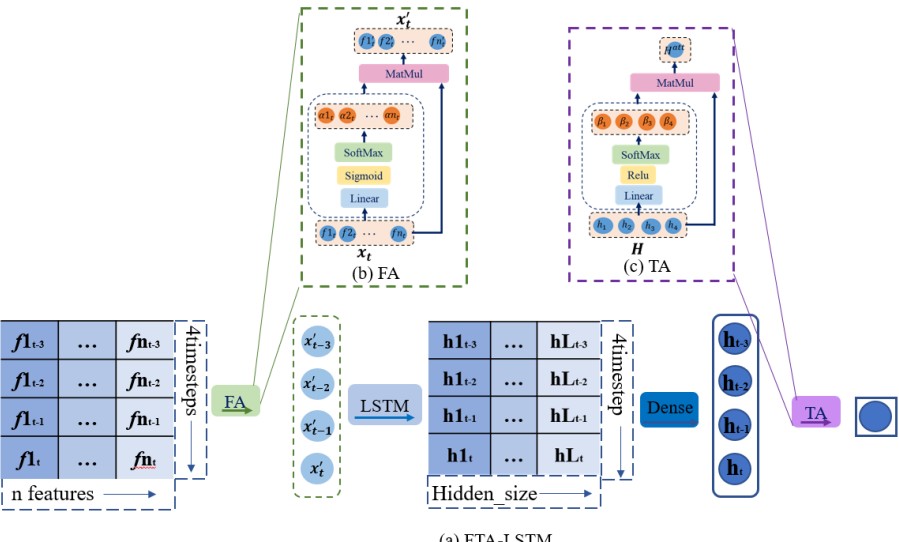

**Figure 6.** Framework of the proposed FA-LSTM hybrid models (a), the feature attention mechanism (FA) (b), and the temporal attention mechanism (TA) (c), inspired by Ding et al.(2020).

**FA-LSTM:**

FA-LSTM applies an attention mechanism to assign weights for distinct features in the input vector. In this study, for soil moisture prediction, the feature attention mechanism in FA-LSTM processes the input vector $I, \{x_{t-3}, ..., x_t\}$, where $x_t = \{f1_t, f2_t, ..., fn_t\}$ and generate the weighted output

$\{\boldsymbol{x}'_{t-3}, \dots, \boldsymbol{x}'_t\}$. Through the attention mechanism, the output $\boldsymbol{x}'_t$ remains the same dimension size as the input $\boldsymbol{x}_t$. The feature attention importance weight $\alpha_t$ and attention mechanism output $\boldsymbol{x}'_t$ are defined as follows:

$$\alpha_t = FA(\boldsymbol{x}_t) \tag{12}$$

$$\boldsymbol{x}'_t = \alpha_t \otimes \boldsymbol{x}_t \tag{13}$$

Fig. 6b also shows the operation of the feature attention mechanism. The FA-LSTM model consists of an LSTM and a feature attention mechanism for input preprocessing, as detailed in Table A1.

**TA-LSTM:**

TA-LSTM utilizes the temporal attention mechanism to weigh the importance of LSTM output vectors across time steps. This enables the model to concentrate on the most relevant hidden states, potentially enhancing its performance on tasks that involve temporal modeling. The temporal attention mechanism is shown in Fig. 6c. In our work, the output vector $\boldsymbol{H}^{att}$, which is obtained through the temporal attention mechanism in TA-LSTM, is the weighted sum of all states in $\boldsymbol{H}, \{\boldsymbol{h}_{t-3}, \boldsymbol{h}_{t-2}, \boldsymbol{h}_{t-1}, \boldsymbol{h}_t\}$. The temporal attention weight $\beta$ and attention mechanism output $\boldsymbol{H}^{att}$ can be defined as:

$$\beta = TA(\boldsymbol{H}) \tag{14}$$

$$\boldsymbol{H}^{att} = \sum_{i=1}^{4} \beta_i \otimes \boldsymbol{h}_i \tag{15}$$

Compared to LSTM, the difference with TA-LSTM lies in the post-processing of the LSTM output. LSTM utilizes the last hidden state output for prediction, while TA-LSTM employs temporal weighting to utilize all hidden state outputs. Table A1 contains the network structure and parameters information.

**FTA-LSTM:**

FTA-LSTM is the model that combines both feature and temporal attention mechanisms, as illustrated in Fig. 6a. It applies the feature attention mechanism before the LSTM layer to assign weights for the input features, and the temporal attention mechanism after the LSTM layer to weigh the importance of the LSTM output vectors of different time steps. The parameters of FTA-LSTM can be found in Table A1.

**3.3 GAN-LSTM**

GANs (Goodfellow et al., 2014) comprise a generator and a discriminator. The generator is designed
to generate predictions that are similar to the truth, while the discriminator tries to distinguish between
the truth and the predictions. The unique network structure and adversarial training of GANs make them
highly effective in various fields, particularly in dealing with fuzzy prediction (Jing et al., 2019). Thus,
GANs offer a promising way to predict soil moisture, potentially leading to accurate results in real
situations. For predicting soil moisture, the GAN-LSTM model is used, where the generator G employs
an LSTM model capable of processing time series data, and the discriminator D uses a single-layer
feedforward neural network, similar to the work of Li et al. (2020). Alternating adversarial training is
performed between G and D, meaning that one of them is trained while keeping the other one fixed. The
structure and training strategies of GAN-LSTM are shown in Fig. 7.

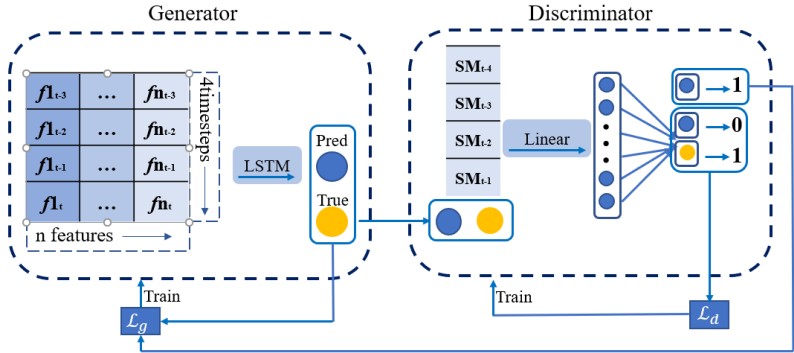

**Figure 7.** The framework of the proposed GAN-LSTM model.

The training objective of the discriminator D is to distinguish between predictions generated by the
generator G and the ground truth, by minimizing the loss function $\mathcal{L}_d$. The binary cross-entropy loss is
utilized as the similarity evaluation metric, with the objective of training D to output 1 when presented
with ground truth as input and 0 when presented with predictions as input:

$$\mathcal{L}_d = \mathcal{L}_{bce}(d([SM_{t-4}, .., SM_{t-1}, y_t]), 1) + \mathcal{L}_{bce}(d([SM_{t-4}, .., SM_{t-1}, \hat{y}_t]), 0) \tag{16}$$

where $\mathcal{L}_{bce}$ is the binary cross-entropy loss, which is defined as:

$$\mathcal{L}_{bce}(\hat{p}, p) = -plog(\hat{p}) - (1 - p)log(1 - \hat{p}) \tag{17}$$

where $p$ denotes the label (0 or 1) and $\hat{p}$ denotes the logit value between 0 and 1.

For generator G, there are two training objectives: first, to generate soil moisture dynamics predictions
that are accurate and consistent with the truth, which is achieved by minimizing the fitting error of the

soil moisture content data, denoted as $\mathcal{L}_{mse}$. Second, to deceive D, which is achieved by minimizing the

binary cross-entropy loss $\mathcal{L}_{bce}$ between the predictions and the truth in D. The output of D should be

close to 1 when inputting the G predictions into D, ensuring that the prediction is close to the truth.

Therefore, we train G by minimizing the following loss function $\mathcal{L}_g$:

$$\mathcal{L}_g = \mathcal{L}_{mse}(y_t, \hat{y}_t) + \lambda_{bce}\mathcal{L}_{bce}(d([SM_{t-4}, .., SM_{t-1}, \hat{y}_t]), 1) \tag{18}$$

where $\lambda_{bce}$ is the hyperparameter that controls the importance of the second term. Here we determine

$\lambda_{bce}$ to be $1 \times 10^{-7}$ through manual testing. For adversarial training in our GAN-LSTM, the parameter

update ratio of G and D in the model is 3:1, that is, every time G is updated (the learning rate is set to

0.0005), D will be updated 3 times (the learning rate is set to 0.001). The network structure parameters

of GAN-LSTM are recorded in Table A1.

## 4. Results and Discussions

This study evaluates the performance of 3 machine learning methods and 10 deep learning models in

predicting soil moisture at 10 sites and 5 depths. To evaluate the model's ability to predict over time

series, we examined forecasts for 1, 3, and 7 days ahead. When making predictions longer than 1 day,

we adopted iterative predictions. The generated soil moisture data for the first day, along with the

corresponding observed meteorological data and historical three-day data reconstruct the new four-day

input, which is used to predict soil water for the second day. Two standard metrics, $R^2$ and root mean

square error (RMSE) are used to evaluate the performance of the models. $R^2$ represents how well the

model captures the variability in data, while RMSE measures the accuracy of the model's predictions.

These metrics are calculated as follows:

$$R^2 = 1 - \frac{\sum_{i=1}^{N}(y_i - \hat{y}_i)^2}{\sum_{i=1}^{N}(y_i - \bar{y}_i)^2} \tag{19}$$

$$RMSE = \sqrt{\frac{\sum_{i=1}^{N}(y_i - \hat{y}_i)^2}{N}} \tag{20}$$

where $y_i$ denotes the ground truth; $\hat{y}_i$ denotes the model prediction, $\bar{y}_i$ denotes the mean of the

ground truth, and N denotes the sample size.

The collected data in Section 2 is split into training, validation, and test sets in a 6:2:2 ratio in time

order. The training set is used to train the models with a learning rate of 0.001 unless stated otherwise.

We train the deep learning models for at least 1500 epochs, with a batch size of 50. In each epoch, 20

batches are used for training. The validation set is employed to determine whether the deep learning

model should be updated. If the trained model performs worse on the validation set compared to the

previous model, the previous model is retained. Finally, the test set is utilized to evaluate and compare

the accuracy of the trained models. To ensure statistical robustness, each final result is obtained by

averaging the outcomes of 25 repetitions of the training process.

**4.1 Comparisons of Machine Learning and LSTM**

This section compares the machine learning models with the deep learning model, represented by

LSTM. Table 2 summarizes the $R^2$ between the soil moisture predictions of the three machine learning

methods and the ground truth at ten sites and five depths for the following 1, 3, and 7 days. The symbol

(-) signifies an extremely poor $R^2$ result. The results show that all three methods perform well on short-

term (1-3 days) soil moisture forecasts, but their performance tends to diverge when predicting at the

lead time of 7 days. Among these three, RF is the most stable and best-performing model.

Fig. 8(a-e) compares the average RMSE of the soil moisture predictions of the machine learning

models and LSTM at different depths for 1, 3, and 7 days ahead across 30 sites. It reveals that LSTM

outperforms the three machine learning models in terms of prediction accuracy and stability, which

suggests that deep learning has a better capability of processing time series data for soil moisture

dynamics simulation than traditional machine learning.

Machine learning models are limited in handling inputs from multiple time steps when processing time

series data. Therefore, while they exhibit proficiency in short-term predictions, they may not perform

well in long-term prediction tasks and demonstrate comparatively lower accuracy and stability than deep

learning models. Nevertheless, a notable advantage of machine learning models is that they require little

training time, enabling rapid deployment, which incurs lower computational costs compared to deep

learning models.

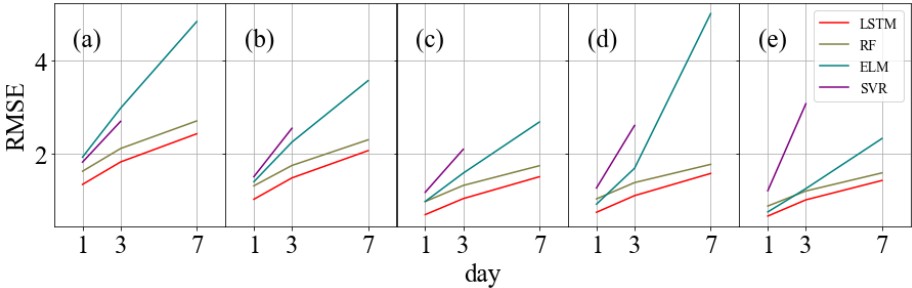

**Figure 8.** Top: RMSE comparisons between RF, ELM, SVR, and LSTM at the Cape site at 5 depths: 0.05m(a),
0.10m(b), 0.20m(c), 0.50m(d), 1.00m(e).

**Table 2.** The values of $R^2$ between the predictions (1, 3, and 7 days) of RF, ELM, and SVR and the ground truth
for ten sites at five depths.

| depth/m | RF $R^2$ | | | ELM $R^2$ | | | SVR $R^2$ | | |
|---|---|---|---|---|---|---|---|---|---|
| | 1d | 3d | 7d | 1d | 3d | 7d | 1d | 3d | 7d |
| 0.05 | **0.924** | **0.874** | **0.797** | 0.889 | 0.735 | (-) | 0.910 | 0.816 | (-) |
| 0.10 | **0.930** | **0.886** | **0.815** | 0.922 | 0.823 | 0.459 | 0.920 | 0.814 | (-) |
| 0.20 | **0.929** | **0.891** | **0.832** | 0.927 | 0.809 | 0.361 | 0.914 | 0.759 | (-) |
| 0.50 | 0.898 | **0.814** | **0.725** | **0.914** | 0.503 | (-) | 0.860 | 0.528 | (-) |
| 1.00 | 0.903 | **0.818** | **0.671** | **0.909** | 0.805 | 0.170 | 0.768 | (-) | (-) |

### 4.2 Comparisons of 1D-CNN, LSTM and Transformer

In this section, we conduct a comparative analysis of three basic deep learning networks. We evaluate
their prediction performance by assessing both prediction accuracy and computational costs. The values
of $R^2$ between the soil moisture predictions generated by the three models and the ground truth across
ten sites and five depths are presented in Table 3. Additionally, Fig. 9(a-e) displays the average RMSE
for soil moisture predictions of 30 sites.

The results reveal that the LSTM model achieves the highest prediction accuracy, followed by the 1D-
CNN model and subsequently the Transformer model. Notably, LSTM and Transformer are more stable
when making long-term or deep soil moisture predictions, while 1D-CNN is better suited for short-term
and shallow prediction tasks. This aligns with the inherent characteristics of the three models. In essence,
LSTM is designed to model temporal dependencies in sequence data, emphasizing global features.
Transformer operates by modeling relationships in input time series without iterations and highlights
important features by self-attention weighting. These characteristics prevent overfitting in the LSTM and
Transformer, resulting in stability in long-term predictions. In contrast, 1D-CNN excels at extracting and

expressing local features, which facilitates it to capture the connections between subtle feature changes and their corresponding outcomes. This capability allows for adaptation to shallow soil moisture prediction tasks with significant variations.

Fig. 9g shows the training epochs required for each model, while Fig. 9h illustrates the time taken for 100 epochs. The 1D-CNN demonstrates the fastest training speed and achieves early convergence.
Conversely, LSTM shows slower training speed attributed to its iterations. The Transformer trains quickly but converges at a slower pace than LSTM, resulting in a similar total training time. In summary, although 1D-CNN offers the lowest computational costs, LSTM has been proven to be the most appropriate for soil moisture prediction tasks among the three with the highest accuracy.

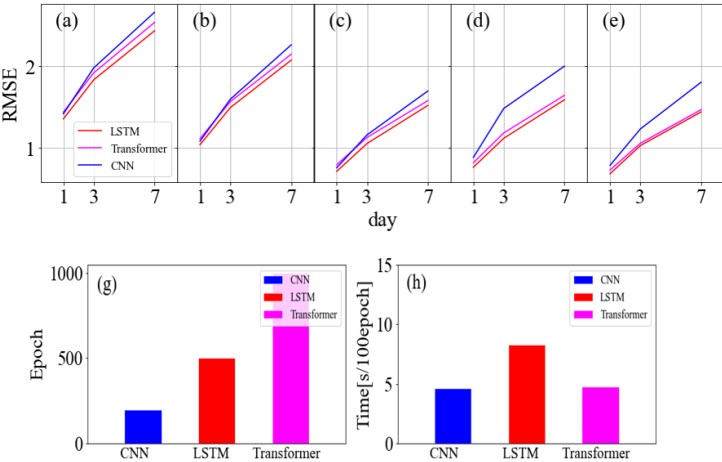


**Figure 9.** Top: Average RMSE comparisons between CNN, LSTM, and Transformer at 5 depths: 0.05m(a), 0.10m(b), 0.20m(c), 0.50m(d), 1.00m(e). Bottom: comparisons of the training epoch (g) and training time (h) for three models.

**Table 3**. The values of $R^2$ between the predictions (1, 3, and 7 days) generated by CNN, LSTM, and Transformer, and the ground truth across ten sites at five depths.

| depth/m | CNN $R^2$ | | | LSTM $R^2$ | | | Transformer $R^2$ | | |
|---|---|---|---|---|---|---|---|---|---|
| | 1d | 3d | 7d | 1d | 3d | 7d | 1d | 3d | 7d |
| 0.05 | 0.939 | 0.884 | 0.793 | **0.943** | **0.895** | **0.816** | 0.933 | 0.886 | 0.805 |
| 0.10 | **0.956** | 0.909 | 0.826 | 0.954 | **0.909** | 0.838 | 0.949 | 0.906 | **0.839** |
| 0.20 | 0.961 | 0.912 | 0.823 | **0.963** | **0.916** | 0.842 | 0.952 | 0.912 | **0.843** |
| 0.50 | 0.909 | 0.702 | 0.532 | **0.937** | **0.873** | **0.749** | 0.917 | 0.840 | 0.716 |
| 1.00 | 0.919 | 0.811 | 0.547 | **0.944** | 0.878 | 0.746 | 0.939 | **0.879** | **0.758** |

### 4.3 Comparisons of CNN and LSTM Hybrid Models

This section compares the three CNN and LSTM hybrid models (LSTM-CNN, CNN-LSTM, and CNN-with-LSTM) across 10 sites in terms of prediction accuracy and computational costs. Table 4 presents the e $R^2$ values between the soil moisture predictions generated by the three hybrid models and the ground truth across ten sites at five depths. It can be observed that the prediction accuracy of these models is comparable, with LSTM-CNN slightly outperforming the others. Moreover, Fig. 10(a-e) shows the average RMSE results of hybrid models and LSTM across 30 research sites, indicating that the hybrid models do not exhibit obvious advantages over the standard LSTM.

Specifically, the three models are hybrids of CNN and LSTM with varying incorporation degrees. According to their combination ways, we can infer that the models excel in handling different types of data and place different emphases on data characteristics. The CNN-LSTM appears to prioritize local features and model long-distance dependencies, while LSTM-CNN focuses on global features and context information. CNN-with-LSTM simultaneously considers both local features and temporal information for predictions. These integrations increase the complexity and enhance the expression capacities of models, but their applications should depend on the input data and prediction task. In the case of soil moisture prediction, the benefits of this combination approach are not significant.

Fig. 10g and Fig. 10h display the computational costs of the three hybrid models. It is evident that the CNN-LSTM shows the fastest training speed and the lowest computational costs, owing to its convolution layers for input data pre-processing. Besides, the computational costs of LSTM-CNN are higher than CNN-with-LSTM. Overall, compared to LSTM and 1D-CNN, we could draw the conclusion that the hybrid models have limited practical values in soil moisture prediction.

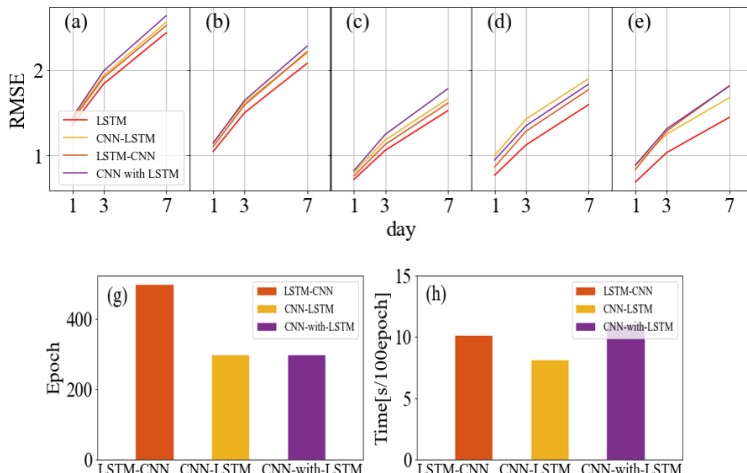

**Figure 10.** Top: Average RMSE comparisons between LSTM-CNN, CNN-LSTM and CNN-with-LSTM at 5 depths: 0.05m(a), 0.10m(b), 0.20m(c), 0.50m(d), 1.00m(e). Bottom: comparisons of the training epoch (g) and training time (h) for three models.

**Table 4**. The values of $R^2$ between the predictions (1, 3, and 7 days) of LSTM-CNN, CNN-LSTM, and CNN-with-LSTM and the ground truth for ten sites at five depths

| depth/m | LSTM-CNN $R^2$ | | | CNN-LSTM $R^2$ | | | CNN-with-LSTM $R^2$ | | |
|---------|-------|-------|-------|-------|-------|-------|-------|-------|-------|
|         | 1d    | 3d    | 7d    | 1d    | 3d    | 7d    | 1d    | 3d    | 7d    |
| 0.05    | **0.939** | **0.889** | **0.809** | 0.936 | 0.885 | 0.800 | 0.936 | 0.880 | 0.792 |
| 0.10    | 0.950 | **0.901** | 0.820 | 0.943 | 0.895 | **0.821** | **0.951** | 0.899 | 0.810 |
| 0.20    | **0.959** | **0.906** | **0.822** | 0.952 | 0.899 | 0.816 | 0.950 | 0.891 | 0.795 |
| 0.50    | **0.916** | **0.814** | **0.683** | 0.867 | 0.715 | 0.546 | 0.886 | 0.782 | 0.644 |
| 1.00    | 0.908 | 0.788 | 0.546 | **0.908** | **0.821** | **0.651** | 0.897 | 0.787 | 0.575 |

## 4.4 Comparisons of Attention Mechanisms and LSTM Hybrid Models

To investigate the impact of different attention mechanisms on models, this section compares these three models: FA-LSTM, TA-LSTM, and FTA-LSTM. Fig. 11(a-e) displays the average RMSE values of the soil moisture predictions for 1, 3, and 7 days ahead generated by these three models and the standard LSTM at 30 sites. Table 5 records the values of $R^2$ between the soil moisture predictions of three models and the ground truth across ten sites and five depths.

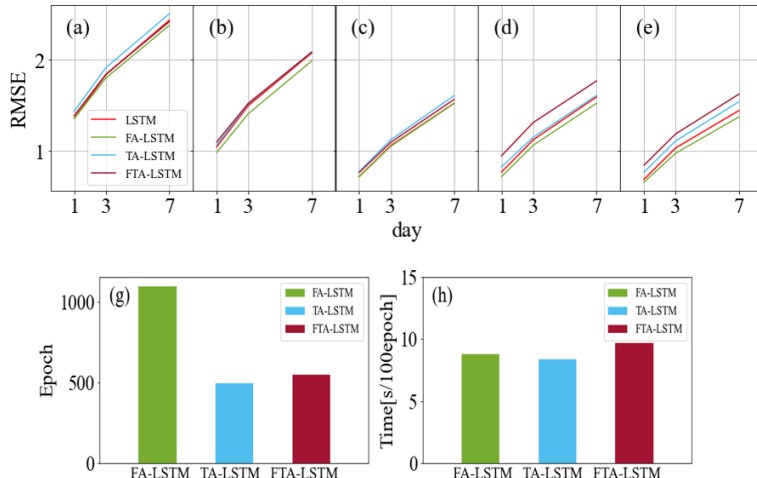

**Figure 11.** Top: Average RMSE comparisons between FA-LSTM, TA-LSTM, and FTA-LSTM at 5 depths: 0.05m(a), 0.10m(b), 0.20m(c), 0.50m(d), 1.00m(e). Bottom: comparisons of the training epoch (g) and training time (h) for three models.

**Table 5**. The values of $R^2$ between the predictions (1, 3, and 7 days) of FA-LSTM, TA-LSTM, and FTA-LSTM and the ground truth for ten sites at five depths.

| depth/m | FA-LSTM $R^2$ | | | TA-LSTM $R^2$ | | | FTA-LSTM $R^2$ | | |
|---|---|---|---|---|---|---|---|---|---|
| | 1d | 3d | 7d | 1d | 3d | 7d | 1d | 3d | 7d |
| 0.05 | **0.944** | **0.902** | **0.827** | 0.937 | 0.888 | 0.809 | 0.942 | 0.897 | 0.823 |
| 0.10 | **0.960** | **0.921** | **0.848** | 0.950 | 0.899 | 0.826 | 0.950 | 0.906 | 0.839 |
| 0.20 | **0.965** | **0.925** | **0.849** | 0.957 | 0.909 | 0.823 | 0.954 | 0.907 | 0.825 |
| 0.50 | **0.949** | **0.881** | **0.770** | 0.923 | 0.869 | 0.745 | 0.870 | 0.773 | 0.653 |
| 1.00 | **0.947** | **0.896** | **0.794** | 0.927 | 0.854 | 0.703 | 0.915 | 0.842 | 0.672 |

Based on the results, the prediction accuracy of the three models ranked from high to low is FA-LSTM, FTA-LSTM, and TA-LSTM in most situations. It can be found that the feature attention mechanism has a stable gain effect on LSTM, potentially because it assigns the appropriate feature importance weights to various influencing factors, especially in deep soil moisture prediction tasks. On the contrary, the improvement of the temporal attention mechanism is not evident and may lead to deterioration. TA-LSTM differs from LSTM in its output post-processing, as it is trained to weigh the LSTM output at each time step to make predictions. The reason why TA-LSTM is worse may be that LSTM already encodes enough past features for predictions in the last hidden state. Moreover, the FTA-LSTM model, which combines both feature and temporal attention mechanisms, is the most complex but not necessarily the

optimal one among the three. From the results, we can also infer the effective feature learning ability of attention mechanisms.

According to Fig. 11(g-h), attention mechanisms introduce some acceptable computational costs. Notably, FA-LSTM requires more training steps to reach convergence. However, despite this computational requirement, we believe that the implementation of FA-LSTM is still advantageous for soil moisture prediction tasks.

Fig. 12 provides visualizations of the input feature importance and temporal importance weights learned by FA-LSTM and TA-LSTM for soil moisture prediction at the AAMU site across 5 depths. The feature importance in Fig. 12 (a-e) shows a reasonable adaptation to the varying depth, demonstrating the effective feature selection capability of attention mechanisms. Moreover, the temporal importance in Fig. 12(f-j) indicates the high utilization of recent temporal features, which is consistent with real situations. This indicates the effective feature learning capacity of attention mechanisms. What's more, these results contribute to a deeper understanding of the utilization mechanisms of feature and temporal information within the model.

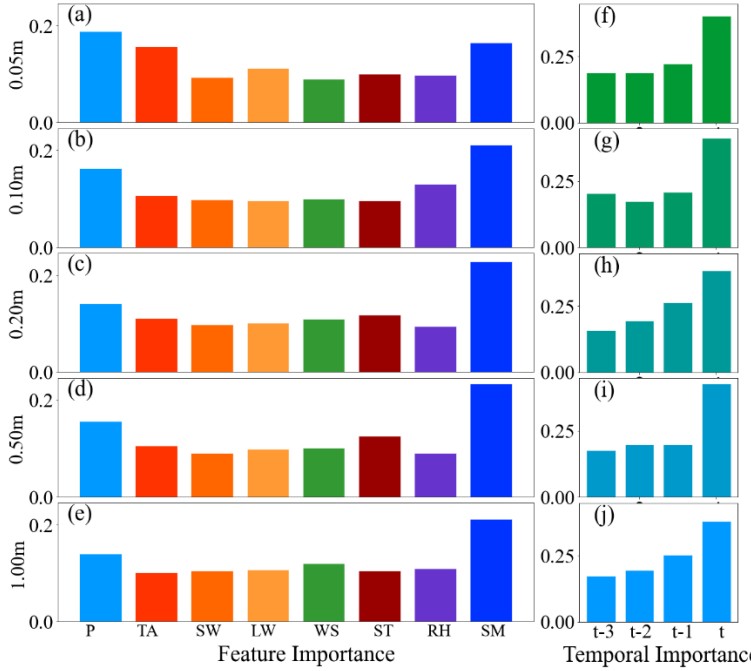

**Figure 12.** Feature importance and temporal importance for soil moisture prediction at the AAMU site across 5 depths.

**4.5 Comparisons of GAN-LSTM and LSTM**

In this section, we evaluate the impact of the GAN structure and adversarial training strategy on the standard LSTM model. LSTM and GAN-LSTM for soil moisture prediction are compared. The $R^2$ values for the following 1, 3, and 7 days across ten sites at different depths are recorded in Table 6. Fig. 13(a-e) shows the RMSE results of LSTM and GAN-LSTM at the Weslaco site.

The results demonstrate that the GAN-LSTM achieves better performance than the standard LSTM in most situations, particularly in long-term prediction tasks (3-7 days). The application of GAN structure and training strategies enhances the prediction accuracy of LSTM. The adversarial training of GAN-LSTM allows the model to not only learn from the data itself but also extract additional information embedded in the data. This helps address performance degradations due to overfitting on data mean

square error. We can regard this training strategy as a general principle to enhance the performance of neural networks. However, the selection of hyperparameters in the loss function of GAN is crucial and currently requires manual adjustments. In future work, adaptive methods can be adopted to automatically adjust the GAN-LSTM loss function to increase training flexibility and prediction accuracy.

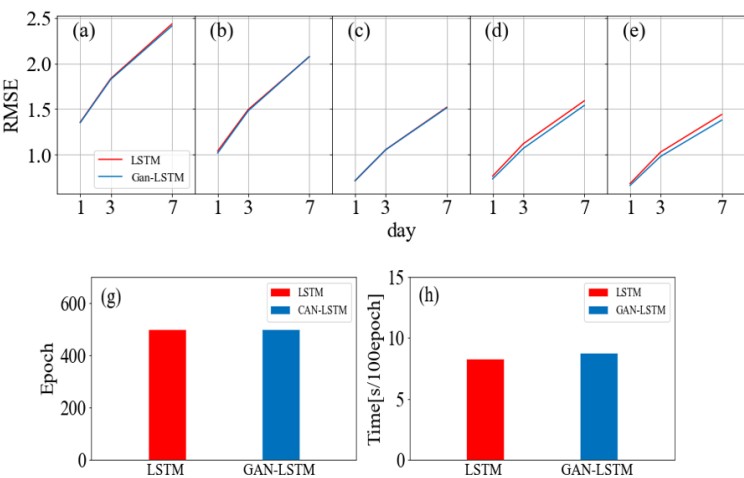

**Figure 13.** Top: Average RMSE comparisons between LSTM and GAN-LSTM at 5 depths: 0.05m(a), 0.10m(b), 0.20m(c), 0.50m(d), 1.00m(e). Bottom: comparisons of the training epoch (g) and training time (h).

**Table 6.** The values of $R^2$ between the predictions (1, 3, and 7 days) generated by LSTM and GAN-LSTM and the

ground truth across ten sites at five depths.

| depth/m | LSTM $R^2$ | | | GAN-LSTM $R^2$ | | |
|---|---|---|---|---|---|---|
| | 1d | 3d | 7d | 1d | 3d | 7d |
| 0.05 | 0.943 | 0.895 | 0.816 | **0.944** | **0.897** | **0.819** |
| 0.10 | 0.954 | 0.909 | **0.838** | **0.956** | **0.910** | 0.838 |
| 0.20 | 0.963 | 0.916 | 0.842 | **0.963** | **0.919** | **0.846** |
| 0.50 | 0.937 | 0.873 | 0.749 | **0.946** | **0.893** | **0.777** |
| 1.00 | 0.944 | 0.878 | 0.746 | **0.948** | **0.896** | **0.793** |

Based on the computational cost comparisons presented in Fig. 13(g-h), both LSTM and GAN-LSTM

exhibit similar computational costs. Consequently, in most scenarios, it is advisable to apply the GAN-

LSTM model to predict soil moisture dynamics. It improves the stability and prediction ability of the

model without imposing a significant increase in computational costs.

**4.6 Visualization Analysis**

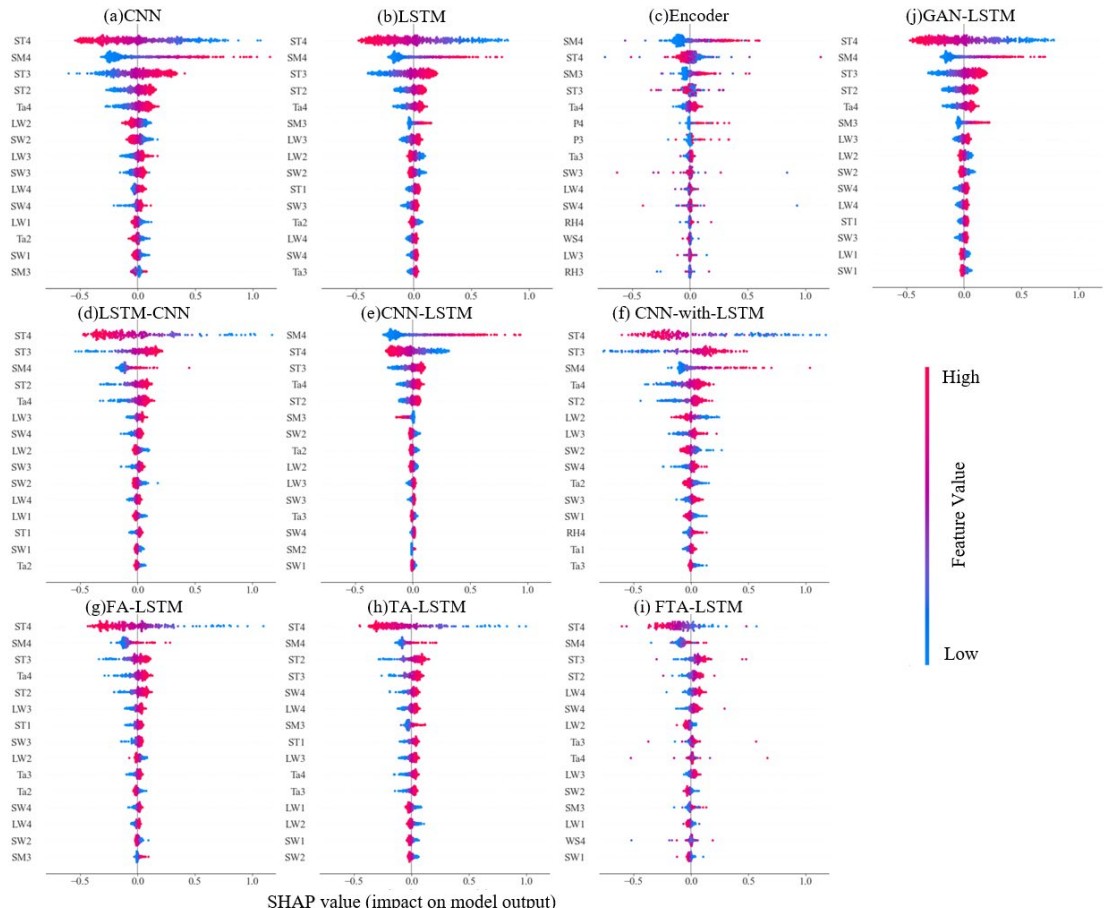

**Figure 14.** SHAP summary plots for ten deep learning models. The samples are from the test set of the Monahans site at 0.0500m.

In this study, we employ the SHAP (Lundberg et al., 2018) to quantify the contributions of input features to investigate the distinct mechanisms of data utilization in different network structures. Brief introductions to SHAP are provided in Appendix B. Fig. 14 illustrates the SHAP summary plots of these ten deep learning models utilizing samples from the test set of the Monahans site. The y-axis represents the input features ranked by importance. Each point shows the Shapley value of a specific feature in a sample, with the color indicating the value of the input feature. The plot clearly shows the identified main influential factors and established correlations between input features and soil moisture by the models. We aim to analyze different ways of using data across various models. For soil moisture predictions, a SHAP analysis visualization for a well-performing model should reflect its emphasis on influential features for improved results in surface soil moisture or short-term predictions. Simultaneously, it can

illustrate the model's capability to avoid overlearning irrelevant features. This avoidance can prevent false correlations that can degrade long-term forecast performance.

Fig. 14(a-c) displays the Shapley values of three basic deep learning models: CNN, LSTM, and Transformer. It can be observed that CNN shows a broader range of Shapley values compared to the others, indicating its greater feature expression capacity. This suggests that CNN focuses more on specific local features, while LSTM emphasizes capturing global features. However, both CNN and LSTM tend to learn incorrect correlations. For instance, the learned positive correlation between the feature ST3 and soil moisture is contrary to the facts. The Transformer model, which aggregates features from all other inputs, appears to perform better in this aspect. Although the Shapley value of the Transformer exhibits the lowest range, the important features identified are derived from the recent input time series, which aligns better with real situations. This reflects the effective feature learning ability of attention mechanisms. Overall, each of these models—CNN, LSTM, and Transformer—possesses unique advantages in terms of data utilization. Notably, LSTM aligns most consistently with the above criteria.

Fig. 14(d-f) compares the hybrid models of CNNs and LSTMs. The CNN-LSTM keeps high Shapley values in important features while showing minimal response to the others. This suggests that CNN-LSTM tends to sequentially process the extracted crucial features, enabling itself to effectively capture both local data features and long-range dependencies, resembling more the CNN. LSTM-CNN shows similar Shapley values to the LSTM. By employing CNN to extract sequential modeling features, LSTM-CNN emphases more on global features, resembling the characteristics of the LSTM. The Shapley value of the CNN-with-LSTM is the highest, displaying a heightened sensitivity to feature perturbations. This can be attributed to the repeated utilization of features in parallel networks. These three models represent different degrees of fusion between CNNs and LSTMs, and the hybrid architecture design depends on the specific task requirements and data characteristics.

In the case of hybrid models that integrate attention mechanisms with LSTM, FA-LSTM, TA-LSTM, and FTA-LSTM, their Shapley values in Fig. 14(g-i) are found to differ slightly from that of LSTM. Considering the attention importance analysis discussed in section 4.4, we can infer that the attention mechanisms introduce slight adjustments to the time and feature attributions on the basis of the LSTM. Fig. 14j also presents the Shapley value of GAN-LSTM. Through the Shapley value, we can infer that the GAN-LSTM model introduces slight modifications during adversarial training, influencing some

feature contributions to improve the prediction accuracy of the LSTM model. This demonstrates that

adversarial training strategies contribute to the refinement and enhancement of models.

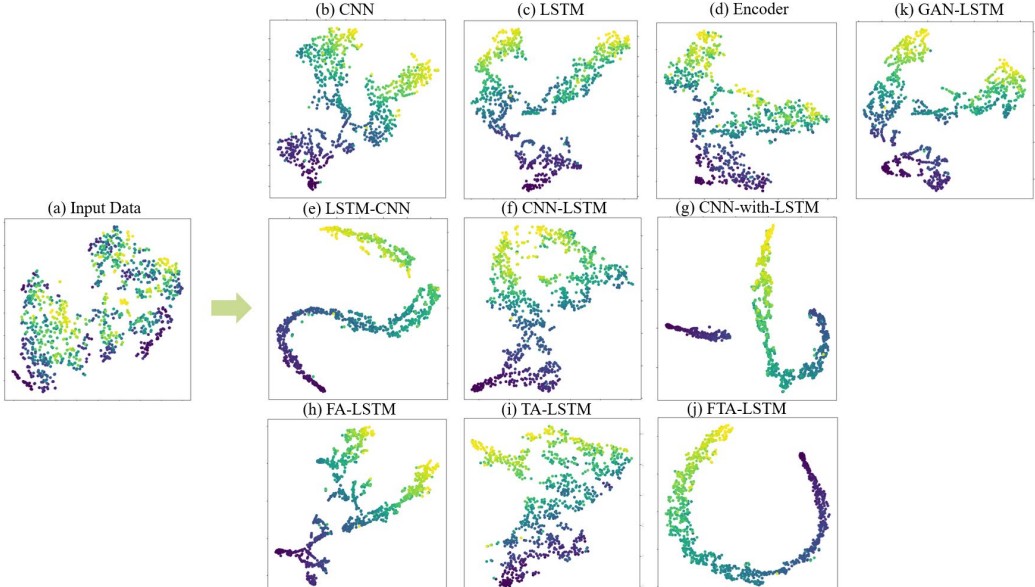

**Figure 15.** The t-SNE visualizations of the original input data (a) and encoded hidden states of 10 models (b-k) obtained from the test set of the LittleRiver site. The colors of the points indicate the corresponding soil moisture content values.


Besides, t-SNE (Van der Maaten and Hinton, 2008), a dimension reduction and visualization method is employed to discover the structure and patterns in the high-dimensional data. When mapping data onto a two-dimensional space, t-SNE retains the relative distance relationships between the original data points, ensuring that similar samples are mapped closer to each other. The details of t-SNE can be found

in Appendix B. Fig. 15 presents the t-SNE visualizations of the input data and the last encoded hidden states from 10 models. The input data in Fig. 15a denotes the flattened form of the four days' inputs $I$. When conducting t-SNE visualizations, the x and y axes make no sense. Only the relative distance between sample points matters. The color of each point corresponds to the soil moisture content value. It is evident that through training, the low-dimensional embeddings of the encoded hidden states

gradually transition from an initially irregular pattern to a more structured shape. However, the visualization shapes vary across the different models. For a soil moisture prediction regression task, we discover that in t-SNE visualizations of models with great forecasting capacity, the sample points can be arranged vertically from light to dark in color, such as the Fig. 15h. Additionally, these visualizations enable us to discern the impact of the attention mechanism and adversarial training on LSTM in Fig.

15(k-h), ultimately leading to enhanced accuracy. However, LSTM-CNN, CNN-with-LSTM, and FTA-LSTM exhibit distinct clustering patterns in their embedding plots, rather than a vertical arrangement. This reflects their advanced data processing capabilities but is less beneficial for soil water prediction tasks. Generally, from the t-SNE visualizations, it can be summarized that different deep learning models capture distinct intrinsic characteristics of input data and encode them into various vectors

for making predictions. To gain a more comprehensive understanding of their differences, further research is warranted in the future.

## 5. Conclusions

In this research, we have conducted a comprehensive analysis of traditional machine learning models

and various deep learning models for soil moisture predictions across different sites at 5 depths. Based on our comparisons of these models, we draw the following conclusions:

In traditional machine learning, RF seems to be the most stable method in soil moisture prediction tasks. However, deep learning models have been found to possess stronger capabilities in processing time series data for better predictions. Among the three basic deep learning models, LSTM demonstrates a

high level of accuracy because of its temporal information modeling capability, while 1D-CNN exhibits the lowest computational cost. Transformer also shows stable long-term forecasting ability. When considering the hybrid models, three combinations of CNN and LSTM did not enhance the prediction abilities in this task. Despite the attractiveness of hybridizing the benefits of CNN and LSTM, the results did not find notable advantages in soil moisture prediction in terms of accuracy and computational costs.

However, the feature attention mechanism has a constant positive effect on LSTM, while temporal attention mechanisms have little significance. In addition, combining generative adversarial network structures and training strategies into LSTM models (GAN-LSTM) has been found to improve prediction accuracy, especially in long-term predictions. To summarize, FA-LSTM and GAN-LSTM are found to be the most stable and effective models for soil moisture prediction. Furthermore, this study attempts to

provide a thorough analysis of models' performances and advance the understanding of machine learning in soil moisture prediction. Through the Shapley analysis, we can infer the different data utilization ways of the 10 models. Besides, the t-SNE visualizations illustrate the varying encoding capabilities in different models.

The results emphasize the importance of appropriate and effective neural network structure design for a given task. For soil moisture prediction, several principles of effective network design can be concluded. Firstly, leveraging the temporal modeling capability of LSTM is well-suited for soil moisture forecasting. Secondly, incorporating attention mechanisms properly facilitates efficient feature learning. The feature selection capability of attention mechanisms has been proven through the performance of the Transformer and the attention mechanisms and LSTM hybrid models. Lastly, applying special GAN structures and adversarial training strategies in models helps extract additional information embedded within data, which also potential for better soil moisture dynamics simulation.

This study provides a reference and lays the groundwork for the development of specialized deep learning models for soil moisture dynamics simulation. However, although data-driven models have shown satisfactory performance, they cannot make long-term predictions precisely due to their lack of physical laws. In the future, the integration of known physical laws with deep learning models will become a promising research direction for soil moisture dynamics simulation.

**ACKNOWLEDGEMENTS**

This work was supported by the National Key Research and Development Program of China (2021YFC3201203), the Priority Research and Development Projects for Ningxia (Grant No.2021BBF02027), and the National Natural Science Foundation of China (Grant 51979200 and 52179038).

**CODE/DATA AVAILABILITY**

The data and codes used in this paper are available on the website (https://doi.org/10.5281/zenodo.10060492).

**AUTHOR CONTRIBUTION**

**Yanling Wang:** Conceptualization, Methodology, Software, Writing- Original draft preparation. **Liangsheng Shi:** Writing-Reviewing and Editing, Supervision. **Yaan Hu:** Writing-Reviewing and Editing. **Xiaolong Hu:** Methodology, Writing-Reviewing and Editing. **Wenxiang Song:** Methodology, Writing-Reviewing and Editing. **Lijun Wang:** Writing-Reviewing and Editing.

**COMPETING INTERESTS**

The contact author has declared that none of the authors has any competing interests.

## Appendix A. Parameters Used in machine learning and deep learning models

**Table A1.** Parameters settings of the deep learning models.

| Network Type | Layers | Kernel_size | Hidden_size (L) | Activation function |
|---|---|---|---|---|
| 1D-CNN | Convolution | 2 | 32 | Tanh |
| | Convolution | 2 | 64 | Tanh |
| | Flatten | | | |
| | Fully-connected | | 1 | Tanh |
| CNN-LSTM | Convolutional | 2 | 32 | Tanh |
| | Convolutional | 2 | 64 | Tanh |
| | LSTM | | 16 | Sigmoid, Tanh |
| LSTM -CNN | LSTM | | 16 | Sigmoid, Tanh |
| | Convolutional | 3 | 32 | Tanh |
| | Convolutional | 3 | 64 | Tanh |
| | Flatten | | | |
| | Fully-connected | | 1 | Tanh |
| CNN-with-LSTM | CNN | | | |
| | Convolutional | 3 | 32 | Tanh |
| | Convolutional | 3 | 64 | Tanh |
| | Flatten | | | |
| | Fully-connected | | 1 | Tanh |
| | LSTM | | | |
| | LSTM | | 16 | Sigmoid, Tanh |
| | Fully-connected | | 1 | Tanh |
| | CNN-with-LSTM | | | |
| | CONCAT | | | |
| | Fully-connected | | 10 | Tanh |
| | Fully-connected | | 1 | Tanh |
| FA-LSTM | F-Attention | | 8 | Sigmoid |
| | LSTM | | 16 | Sigmoid, Tanh |
| TA-LSTM | LSTM | | 16 | Sigmoid |
| | T-Attention | | 8 | Relu |

| Network Type | Layers | Kernel_size | Hidden_size (L) | Activation function |
|---|---|---|---|---|
| FTA-LSTM | F-Attention | | 8 | Sigmoid |
| | LSTM | | 16 | Sigmoid, Tanh |
| | T-Attention | | 8 | Relu |
| | Fully-connected | | 1 | Tanh |
| GAN-LSTM | Generator | | | |
| | LSTM | | 16 | Sigmoid, Tanh |
| | Fully-connected | | 1 | Tanh |
| | Discriminator | | | |
| | Fully-connected | | 1 | Sigmoid |

RF: the default parameter values in RandomForestRegressor of the sklearn library

SVR: C=1.0, ε=0.1, kernel γ = 'poly'

ELM: Hidden_size (L) =20

LSTM: num_layers = 2, Hidden_size(L) = 16

Transformer: d_k=d_v=4, d_model=feature=8, d_ff=20, n_heads=1

## Appendix B. Shapley additive explanations (SHAP)

SHAP (Lundberg et al., 2018) is a game theoretic approach to explain the output of machine learning models. It measures the impact of the input feature on the prediction of an individual sample. SHAP employs the additive feature attribution method to provide a specific explanation:

$$f(x) = g(x') = \phi_0 + \sum_{i=1}^{M} \phi_i x' \tag{B1}$$

where $f(x)$ denotes the original model, $g(x)$ represents the explanation model with simplified input $x', x' \in \{0,1\}^M$, M is the number of input features; through a mapping function, $x = h_x(x')$; $\phi_i$ denotes the feature attribution of feature $i$. The explanation model $g$ has a unique solution:

$$\phi_i(f, x) = \sum_{z' \subseteq x'} \frac{|z'|! \, (M - |z'| - 1)!}{M!} [f_x(z') - f_x(z'\backslash i)] \tag{B2}$$

where $|z'|$ is the non-zero entry number in $z'$, $z' \subseteq x'$; $f(x') = f(h_x(z')) = E[f(z)|z_s]$, S denotes the non-zero indexes set in $z'$.

## Appendix C. The t-Distributed Stochastic Neighbor Embedding (t-SNE) visualization

The t-SNE is a nonlinear dimension reduction technique that assumes the presence of a low-dimensional nonlinear manifold within the high-dimensional data. Its primary task is to bring similar neighboring points close together in the low-dimensional representation. The working process of t-SNE

can be divided into several steps: calculate the similarity between data points in high-dimensional space,

and then calculate the corresponding probability of points in low-dimensional space. The similarity of

points is calculated as conditional probability. If interested, more information can be found in the work

of Van der Maaten and Hinton (2008). The following is the formula for calculating the similarity $P_{ij}$

and probability $q_{ij}$ of the points:

The similarity between data points in high-dimensional space:

$$P_{ij} = \left(P_{j|i} + P_{i|j}\right)/2N \tag{C1}$$

Corresponding probability of points in low-dimensional space:

$$q_{ij} = \frac{\left(1 + \left\|y_i - y_j\right\|^2\right)^{-1}}{\sum_{k \neq l}(1 + \left\|y_k - y_l\right\|^2)^{-1}} \tag{C2}$$

where $P_{i|j}$ denotes the conditional probability of point $i$ picking point $j$ as its neighbor if neighbors

are chosen according to their probability density under a Gaussian distribution centered at $i$, and $N$

denotes the data points number. $y_i$ denotes the low-dimensional representation of point $i$, and

$\left\|y_i - y_j\right\|$ denotes the Euclidean distance between $y_i$ and $y_j$.

## Appendix D. Detailed meteorological information of the research sites

The soil moisture time series data and detailed meteorological information are recorded in this

Appendix.

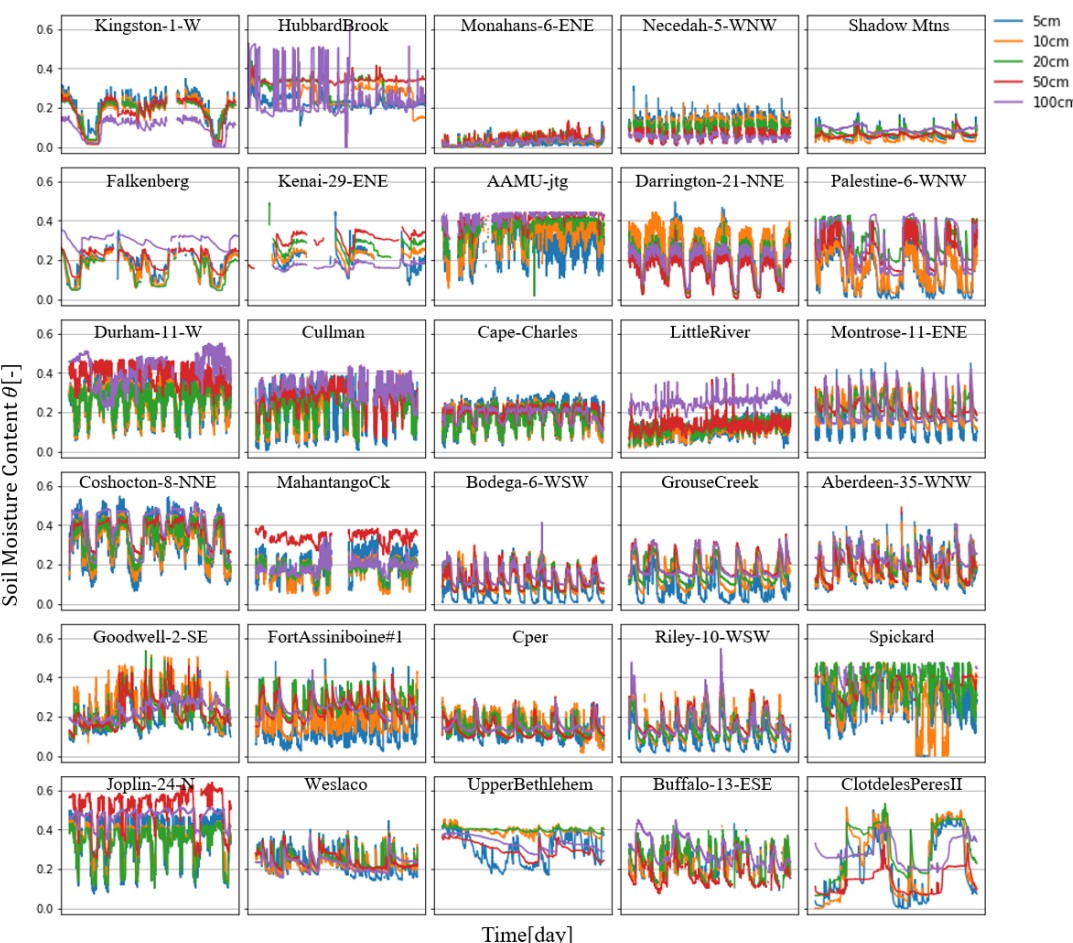

**Figure D1.** Soil moisture content time series data at various depths of thirty sites.

 **Table D2.** Statistical results of P, and TA at 30 station sites

| | | Min | Max | Mean | Std | Training set | Validation | Test set |
|---|---|---|---|---|---|---|---|---|
| Kingston-1-W | P(mm) | 0 | 144.5 | 3.37 | 9.40 | 2012.01.01- 2019.01.26 | 2019-01-26- 2021.06.05 | 2021.06.05- 2023.10.14 |
| | TA(°C) | -18.00 | 28.28 | 10.71 | 9.02 | | | |
| HubbardBrook | P(mm) | 0 | 115.57 | 3.10 | 8.32 | 2003.01.01- 2014.06.02 | 2014.06.02- 2018.03.23 | 2018.03.23- 2022.01.11 |
| | TA(°C) | -25.73 | 27.13 | 7.16 | 10.22 | | | |
| Monahans-6-ENE | P(mm) | 0 | 80.6 | 0.85 | 4.60 | 2010.04.21- 2017.08.25 | 2017.08.25- 2020.02.05 | 2020.02.05- 2022.07.19 |
| | TA(°C) | -12.78 | 36.53 | 19.18 | 8.86 | | | |
| Necedah-5-WNW | P(mm) | 0 | 127.6 | 2.48 | 7.23 | 2009.10.13- 2017.08.27 | 2017.08.27- 2020.04.11 | 2020.04.11- 2022.11.26 |
| | TA(°C) | -28.87 | 30.47 | 7.92 | 11.69 | | | |
| ShadowMtns | P(mm) | 0 | 762.25 | 1.06 | 20.00 | 2013.07.15- 2015.12.19 | 2015.12.19- 2016.10.10 | 2016.10.10- 2017.08.02 |
| | TA(°C) | -2.67 | 35.98 | 17.97 | 8.48 | | | |
| Falkenberg | P(mm) | 0 | 35.34 | 0.73 | 1.95 | 2003.01.17- 2013.07.07 | 2013.07.07- 2017.01.01 | 2017.01.01- 2020.06.30 |
| | TA(°C) | -18.19 | 29.45 | 9.69 | 7.82 | | | |
| Kenai-29-ENE | P(mm) | 0 | 35.7 | 1.25 | 3.08 | 2012.10.04- 2019.05.17 | 2019.05.17- 2021.07.30 | 2021.07.30- 2023.10.14 |
| | TA(°C) | -32.24 | 22.09 | 2.47 | 10.01 | | | |
| AAMU-jtg | P(mm) | 0 | 175.26 | 2.44 | 9.42 | 2010.02.06- 2017.10.07 | 2017.10.07- 2020.04.27 | 2020.04.27- 2022.11.18 |
| | TA(°C) | -10.83 | 31.27 | 16.69 | 8.24 | | | |
| Darrington-21-NNE | P(mm) | 0 | 119.2 | 5.91 | 11.64 | 2013.01.01- 2017.03.13 | 2017.03.13- 2018.08.06 | 2018.08.06- 2019.12.30 |
| | TA(°C) | -7.43 | 24.26 | 9.78 | 6.41 | | | |
| Palestine-6-WNW | P(mm) | 0 | 143.7 | 2.56 | 9.73 | 2009.08.01- 2012.02.18 | 2012.02.18- 2012.12.24 | 2012.12.24- 2013.10.31 |
| | TA(°C) | -6.77 | 34.24 | 19.84 | 8.38 | | | |
| Durham-11-W | P(mm) | 0 | 116.1 | 3.37 | 9.24 | 2011.01.01- 2018.09.07 | 2018.09.07- 2021.03.31 | 2021.03.31- 2023.10.23 |
| | TA(°C) | -10.48 | 29.60 | 15.45 | 8.24 | | | |
| Cullman-NAHRC | P(mm) | 0 | 177.28 | 2.18 | 7.73 | 2006.05.18- 2016.04.19 | 2016.04.19- 2019.08.10 | 2019.08.10- 2022.11.30 |
| | TA(°C) | -10.07 | 30.61 | 16.00 | 8.28 | | | |
| Cape-Charles-5-ENE | P(mm) | 0 | 159.10 | 2.94 | 9.19 | 2011.06.15- 2018.04.13 | 2018.04.13- 2020.07.22 | 2020.07.22- 2022.11.01 |
| | TA(°C) | -10.47 | 32.11 | 15.67 | 8.53 | | | |
| LittleRiver | P(mm) | 0 | 154.68 | 2.95 | 9.62 | 2005.10.18- 2014.04.26 | 2014.04.26- 2017.02.26 | 2017.02.26- 2020.01.01 |
| | TA(°C) | -4.24 | 31.99 | 19.77 | 7.08 | | | |
| Montrose-11-ENE | P(mm) | 0 | 36.9 | 1.32 | 3.55 | 2010.06.21- 2018.06.22 | 2018.06.22- 2021.02.20 | 2021.02.20- 2023.10.23 |
| | TA(°C) | -24.30 | 24.02 | 6.51 | 9.46 | | | |
| Coshocton-8-NNE | P(mm) | 0 | 76.0 | 2.94 | 7.06 | 2009.09.18- 2013.11.30 | 2013.11.30- 2015.04.25 | 2015.04.25- 2016.09.18 |
| | TA(°C) | -20.84 | 29.58 | 10.78 | 10.23 | | | |

| | | Min | Max | Mean | Std | Training set | Validation set | Test set |
|---|---|---|---|---|---|---|---|---|
| MahantangoCk | P(mm) | 0 | 53.09 | 1.12 | 4.89 | 2002.10.17-2006.12.19 | 2006.12.19-2008.05.10 | 2008.05.10-2009.10.01 |
| | TA(°C) | -14.53 | 27.60 | 10.19 | 9.29 | | | |
| Bodega-6-WSW | P(mm) | 0 | 129.6 | 1.86 | 7.07 | 2011.09.18-2018.12.15 | 2018.12.15-2021.05.15 | 2021.05.15-2023.10.14 |
| | TA(°C) | 3.70 | 21.86 | 11.82 | 2.21 | | | |
| GrouseCreek | P(mm) | 0 | 786.38 | 1.24 | 15.09 | 2016.01.01-2020.09.02 | 2020.09.02-2022.03.25 | 2022.03.25-2023.10.15 |
| | TA(°C) | -18.475 | 28.25 | 7.58 | 10.27 | | | |
| Aberdeen-35-WNW | P(mm) | 0 | 70.1 | 1.30 | 4.78 | 2012.01.01-2019.01.31 | 2019.01.31-2021.06.12 | 2021.06.12-2023.10.23 |
| | TA(°C) | -31.07 | 28.97 | 6.13 | 12.81 | | | |
| Goodwell-2-SE | P(mm) | 0 | 72.1 | 1.18 | 4.65 | 2011.08.17-2018.11.09 | 2018.11.09-2021.04.08 | 2021.04.08-2023.09.06 |
| | TA(°C) | -21.85 | 33.65 | 14.01 | 10.24 | | | |
| FortAssiniboine#1 | P(mm) | 0 | 56.90 | 0.82 | 3.51 | 2010.10.01-2017.11.04 | 2017.11.04-2020.03.17 | 2020.03.17-2022.07.29 |
| | TA(°C) | -33.08 | 30.24 | 6.94 | 12.05 | | | |
| Cper | P(mm) | 0 | 177.04 | 1.12 | 6.01 | 2013.09.13-2018.08.01 | 2018.08.01-2020.03.18 | 2020.03.18-2021.11.03 |
| | TA(°C) | -29.26 | 27.45 | 8.16 | 10.50 | | | |
| Riley-10-WSW | P(mm) | 0 | 28.9 | 0.69 | 2.10 | 2011.01.01-2017.07.02 | 2017.07.02-2019.09.02 | 2019.09.02-2021.11.02 |
| | TA(°C) | -20.34 | 29.27 | 7.98 | 9.49 | | | |
| Spickard | P(mm) | 0 | 152.91 | 2.43 | 8.59 | 2010.10.08-2018.01.18 | 2018.01.18-2020.06.22 | 2020.06.22-2022.11.26 |
| | TA(°C) | -22.13 | 32.31 | 11.64 | 11.17 | | | |
| Joplin-24-N | P(mm) | 0 | 138.5 | 3.12 | 9.70 | 2010.01.01-2016.08.06 | 2016.08.06-2018.10.18 | 2018.10.18-2020.12.30 |
| | TA(°C) | -16.72 | 34.26 | 13.88 | 9.92 | | | |
| Weslaco | P(mm) | 0 | 294.89 | 1.65 | 11.66 | 2017.01.01-2019.08.07 | 2019.08.07-2020.06.18 | 2020.06.18-2021.05.01 |
| | TA(°C) | -1.41 | 32.46 | 23.46 | 6.07 | | | |
| UpperBethlehem | P(mm) | 0 | 156.20 | 2.78 | 10.12 | 2008.09.15-2009.09.05 | 2009.09.05-2010.01.01 | 2010.01.01-2010.05.01 |
| | TA(°C) | 21.64 | 28.78 | 25.93 | 1.46 | | | |
| Buffalo-13-ESE | P(mm) | 0 | 92.7 | 1.22 | 4.56 | 2010.08.19-2018.07.15 | 2018.07.15-2021.03.03 | 2021.03.03-2023.10.22 |
| | TA(°C) | -31.14 | 30.74 | 7.18 | 11.77 | | | |
| ClotdelesPeresII | P(mm) | 0 | 33.2 | 0.96 | 3.38 | 2021.07.21-2022.08.14 | 2022.08.14-2022.12.22 | 2022.12.22-2023.05.01 |
| | TA(°C) | -1.34 | 31.36 | 13.09 | 8.28 | | | |

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
