# Peer review of "A comprehensive study of deep learning for soil moisture prediction"

_Hydrology and Earth System Sciences, 2023_

## Community Comment (CC1)

The study conducted a comparison of ten different network structures to assess their predictive abilities and computational costs across various soil textures and depths. The results indicate that Long Short-Term Memory (LSTM), feature attention LSTM (FA-LSTM), and generative adversarial network-based LSTM (GAN-LSTM) are effective in soil moisture forecasting. The study also provides insights into the interpretability of the models and emphasizes the importance of appropriate model design for specific soil moisture prediction tasks. Therefore, this study can serve as a valuable reference for the application of deep learning models in soil water dynamics. Overall, the manuscript is well-organized and easy to follow.

However, there are a few minor issues that the authors should consider. Firstly, it would be beneficial to provide a more detailed description of the representativeness of the sites to avoid potential one-sided conclusions.

Response:

We sincerely thank the reviewer for providing such insightful and detailed comments which have greatly improved the quality of the manuscript. Regarding the reviewer's concern about the data description, we have reorganized Section 2 and added detailed station land cover and meteorology information. Moreover, we have conducted a Pearson correlation analysis for screening input variables.

Response 1:
Table 1 presents the comprehensive details for ten selected sites, sorted from high to low soil permeability. These sites are carefully chosen to illustrate the model's generalization ability, and they encompass ten different soil textures and five distinct land cover types. In addition to site basic meteorology information, Table 2 provides a record of climate data for these selected locations. This data includes minimum, maximum, average, and standard deviation values for air temperature and precipitation. Furthermore, our input data correlation analysis in Figure R1 also demonstrates the variations between the stations.

Table 1. Summary of main characteristics of ten sites.

|  | Sand | Silt | Clay | Land cover | Period | Lat. | Lon. |
|---|---|---|---|---|---|---|---|
| Monahans-6-ENE | 83 | 6 | 11 | Shrubland | 2010-2022 | 31.62 | 102.81 |
| Necedah-5-WNW | 83 | 11 | 6 | Grassland | 2009-2022 | 44.06 | -90.17 |
| Falkenberg | 73 | 21 | 6 | Cropland, rained | 2003-2020 | 52.17 | 14.12 |
| AAMU-jtg | 53 | 22 | 25 | Grassland | 2010-2022 | 34.78 | -86.55 |
| Cullman-NAHRC | 49 | 27 | 24 | Mosaic Cropland | 2006-2022 | 34.20 | -86.80 |
| Cape-Charles-5-ENE | 49 | 27 | 24 | Herbaceous cover | 2011-2022 | 37.29 | -75.93 |
| LittleRiver | 47 | 30 | 23 | Grassland | 2005-2020 | 31.50 | -83.55 |
| Spickard | 35 | 41 | 24 | Grassland | 2010-2022 | 40.25 | -93.72 |
| Weslaco | 34 | 45 | 21 | Cropland, rained | 2017-2021 | 26.16 | -97.96 |
| UpperBethlehem | 32 | 38 | 30 | Herbaceous cover | 2008-2010 | 17.72 | -64.80 |

Table 2. Statistical results of P, and TA at 10 station sites

| | | Min | Max | Mean | Std | Training set | Validation set | Test set |
|---|---|---|---|---|---|---|---|---|
| Monahans-6-ENE | P | 0 | 80.6 | 0.85 | 4.60 | 2010.04.21-2017.08.25 | 2017.08.25-2020.02.05 | 2020.02.05-2022.07.19 |
| | TA | -12.78 | 36.53 | 19.18 | 8.86 | | | |
| Necedah-5-WNW | P | 0 | 127.6 | 2.48 | 7.23 | 2009.10.13-2017.08.27 | 2017.08.27-2020.04.11 | 2020.04.11-2022.11.26 |
| | TA | -28.87 | 30.47 | 7.92 | 11.69 | | | |
| Falkenberg | P | 0 | 35.34 | 0.73 | 1.95 | 2003.01.17-2013.07.07 | 2013.07.07-2017.01.01 | 2017.01.01-2020.06.30 |
| | TA | -18.19 | 29.45 | 9.69 | 7.82 | | | |
| AAMU-jtg | P | 0 | 175.26 | 2.44 | 9.42 | 2010.02.06-2017.10.07 | 2017.10.07-2020.04.27 | 2020.04.27-2022.11.18 |
| | TA | -10.83 | 31.27 | 16.69 | 8.24 | | | |
| Cullman-NAHRC | P | 0 | 177.28 | 2.18 | 7.73 | 2006.05.18-2016.04.19 | 2016.04.19-2019.08.10 | 2019.08.10-2022.11.30 |
| | TA | -10.07 | 30.61 | 16.00 | 8.28 | | | |
| Cape-Charles-5- | P | 0 | 159.10 | 2.94 | 9.19 | 2011.06.15-2018.04.13 | 2018.04.13-2020.07.22 | 2020.07.22-2022.11.01 |
| | TA | -10.47 | 32.11 | 15.67 | 8.53 | | | |
| LittleRiver | P | 0 | 154.68 | 2.95 | 9.62 | 2005.10.18-2014.04.26 | 2014.04.26-2017.02.26 | 2017.02.26-2020.01.01 |
| | TA | -4.24 | 31.99 | 19.77 | 7.08 | | | |
| Spickard | P | 0 | 152.91 | 2.43 | 8.59 | 2010.10.08-2018.01.18 | 2018.01.18-2020.06.22 | 2020.06.22-2022.11.26 |
| | TA | -22.13 | 32.31 | 11.64 | 11.17 | | | |
| Weslaco | P | 0 | 294.89 | 1.65 | 11.66 | 2017.01.01-2019.08.07 | 2019.08.07-2020.06.18 | 2020.06.18-2021.05.01 |
| | TA | -1.41 | 32.46 | 23.46 | 6.07 | | | |
| UpperBethlehem | P | 0 | 156.20 | 2.78 | 10.12 | 2008.09.15-2009.09.05 | 2009.09.05-2010.01.01 | 2010.01.01-2010.05.01 |
| | TA | 21.64 | 28.78 | 25.93 | 1.46 | | | |

Additionally, conducting a sensitivity analysis for the input factors would provide further justification for the input screening of the deep learning models.

Response 2:
In the process of screening input factors, we have carefully selected meteorological inputs based on the precipitation and evapotranspiration calculation. Besides, soil temperature data, along with soil moisture data from the previous day are incorporated to represent the soil condition. Figure R1 displays the Pearson correlation analysis results for input factors at the Cape-Charles and UpperBethlem sites. Notably, the correlation coefficients between soil moisture data and the input data vary greatly with both the station and depth. For instance, while the correlation coefficient between longwave radiation (LW) and soil moisture is low at UpperBethlem, it is significant at Cape-Charles, highlighting the influence of site-specific differences. Although utilizing highly correlated factors as inputs appears to be a logical choice, achieving uniformity across different sites and depths can be a complex task. However, this presents a crucial aspect to explore when evaluating and comparing the performance of models for self-learning screening of significant influencing factors. Therefore, we have chosen to include all eight of these data points as inputs. Figure R2 shows the autocorrelation analysis conducted at 5 soil depths. The autocorrelation coefficients for soil water content at different depths decrease with increasing delay days. The most significant change is observed in the surface layer. As a result, we have opted to use a 4-day delay as our input.

[Figure]

Figure R1. Pearson correlation analysis results among the observed variables of 0.05m and 1.00m at Cape-Charles (a) (b) and UpperBethlem (c) (d) sites.

[Figure]

Figure R2. Autocorrelation analysis results of soil water content with different days delay at Cape-Charles

---

## Author Comment (AC2)

In this paper, the authors explored various network architectures. They not only compared the predictive capabilities of different models over 1, 3, and 7 days but also considered their computational costs. This paper uses Shapley additive explanations (SHAP) and t-SNE visualization to offer valuable insights into deep learning methods for soil moisture forecasting. However, there are still questions in this paper that need further clarification.

Response:

We are very grateful to receive such valuable comments and suggestions on our manuscript from the reviewer. These insights have significantly enhanced the organization and rigor of our work. We have attempted to address all the comments to ensure clearer expressions and more reliable conclusions.

1. Deep learning has made some progress in soil moisture research in recent years. However, the authors have not sufficiently addressed the latest research related to soil moisture and deep learning, such as the work of Sungmin O, Hylke E. Beck, Jiangtao Liu, and Peyman Abbaszadeh, among others.

Response:

Thank you for your suggestions. Our manuscript primarily concentrates on soil moisture prediction research within the context of deep learning using in-situ observations. Owing to the inconsistency of research purposes and applied data types, we could not further compare our work with their research. We will cite their works in our revised introduction to supplement the background on soil moisture machine learning and acknowledge their contributions to the field.

Machine learning has made remarkable progress in soil moisture dynamics simulations, which have demonstrated the capacity for accurate predictions. A notable example of this is that Sungmin O et al. efficiently employed LSTM to interpolate global gridded datasets from in-situ observations (Orth, 2021; Orth et al., 2022); Additionally, when dealing with multi-scale soil moisture data, such as satellite data, Abbaszadeh et al. (Abbaszadeh et al., 2019) introduced an innovative approach. They employed 12 distinct Random Forest models to downscale the daily composite version of SMAP data. Moreover, advancements in model structure have been instrumental in enhancing performance and improving generalization abilities. For instance, Liu et al integrated multi-scale designs into their models (Liu et al., 2022). In addition to pure deep learning models, differentiable, physics-informed machine learning models with a physical foundation have emerged as a noteworthy development. This kind of model systematically integrates physical equations with deep learning, enabling the prediction of untrained variables and processes with high accuracy (Feng et al., 2023).

2. The authors have chosen only ten sites for this study, However, "As of July 2021, the ISMN now contains data from 71 networks and 2842 stations located globally, spanning from 1952 to the present." [https://hess.copernicus.org/articles/25/5749/2021/]. Using less than 1% of these sites hardly represents the global spatial distribution.

Response:

Thank you for your comments. The research objective of our work is to predict soil moisture content at 5 depths across various soil textures and conditions using in-situ observations. Unfortunately, data from most stations are not qualified for this purpose due to the data missing and poor quality. Most previous studies focusing on relevant prediction tasks (Datta and Faroughi, 2023; Liu et al., 2014; Li et al., 2022b, 2020, 2022a; Yu et al., 2021; Gill et al., 2006), adopted 10 in-situ observation

sites for model analysis.

In our case, given the comprehensiveness of soil moisture prediction that our research aims to achieve, we have extended our work to observe the performance of the model in the characteristic patterns of soil moisture changes at different depths. This also enhances the reliability of our study. In response to the reviewer's concerns about the results, we have added data from 20 more observation sites to further consolidate our conclusions.

a). What were the reasons for choosing these ten sites?

Response:

When selecting sites from ISMN, we prioritize evaluating the quantity and quality of available data. The in-situ observations are required to include soil moisture observations at 5 standard depths (0.05m, 0.10m, 0.20m, 0.50m, 1.00m) along with corresponding soil temperature observations. Generally, observations from SCAN and USCRN are more suitable. Subsequently, we randomly select a group of sites that meet our requirements as a preliminary pool for the final site selection. Finally, the research sites are carefully chosen according to the geographical location (dispersed as much as possible), soil textures, and distinct land cover types (diverse as much as possible). We have added an explanation on this in the revised manuscript.

b). Due to the limited number of sites, it is hard to determine the role of CNN.

Response:

Thank you for your comments. We have considered this potential issue. The 30 sites are selected by covering different soil textures and land cover types. In our study, CNN is trained and tested utilizing soil moisture time-series data from one depth at one site every time. Building on this foundation, we conduct a statistical analysis of CNN's training results across various sites and five different depths, providing relatively reliable results.

Besides, the conclusions in Line 430 highlight CNN's great performance in the surface soil or for short-term predictions. And the SHAP visualizations indicate that CNNs are particularly well-suited for learning data with substantial fluctuation, which is consistent with our calculation results.

Notably, our results mainly show the fundamental accuracy and data processing ways for soil moisture dynamics of different models, and should not be regarded as their upper performance limits. There is potential for improvement through further design, which needs further research.

c). If the data quantity were increased, would the results for all models improve or worsen? Does the current conclusion still apply to broader areas?

Response:

In the revised manuscript, we have chosen 20 more sites for model comparisons and evaluations. For each site and depth, we conducted independent training and evaluation using soil moisture time-series data. Figure R1 illustrates the geographical locations of thirty sites. The detailed information on the chosen sites is summarized in Table R1, and the average values of $R^2$ are presented in Table R2. Each result in Table R2 is the average of ten repetitions.

Table R1. Summary of main characteristics of twenty sites.

|  | Sand | Silt | Clay | Land cover | Period | Lat. | Lon. |
|---|---|---|---|---|---|---|---|
| Kingston_1_W | 85 | 10 | 5 | Grassland | 2012-2023 | 41.48 | -71.54 |
| hubard | 85 | 11 | 4 | Treecover | 2003-2022 | 43.93 | -71.72 |
| Shadow Mtns | 79 | 10 | 11 | Shrubover | 43.4 | 35.47 | -115.72 |
| Kenai_29_ENE | 54 | 38 | 8 | Shrubover | 2012-2023 | 60.72 | -150.45 |
| Darrington | 53 | 22 | 25 | Treecover | 2013-2019 | 48.54 | -121.45 |
| Palestine_6_WNW | 49 | 27 | 24 | Grassland | 2009-2013 | 31.78 | -95.72 |
| Durham_11_W | 49 | 27 | 24 | herbaceouscover | 2009-2016 | 40.37 | -81.78 |
| Montrose_11_ENE | 43 | 35 | 22 | Treecover | 2010-2023 | 38.54 | -107.69 |
| Coshocton_8_NNE | 41 | 39 | 20 | Grassland | 2009-2016 | 40.37 | -81.78 |
| Mahan | 41 | 39 | 20 | Cropland | 2002-2021 | 40.67 | -76.67 |
| Bodega-6-WSW | 39 | 38 | 23 | Grassland | 2011-2023 | 38.32 | -123.08 |
| GrouseGreek | 36 | 41 | 23 | Grassland | 2016-2023 | 41.78 | -113.82 |
| Aberdeen_35_WNW | 36 | 41 | 23 | Grassland | 2012-2023 | 45.71 | -99.13 |
| Goodwell | 36 | 41 | 23 | Grassland | 2010-2022 | 36.57 | -101.61 |
| FortAssiniboine#1 | 36 | 41 | 23 | Grassland | 2017-2021 | 48.48 | -109.8 |
| Cper | 36 | 41 | 23 | Grassland | 2013-2021 | 40.82 | -104.71 |
| Riley_10_WSW | 36 | 41 | 23 | Shrubover | 2011-2021 | 43.47 | -119.69 |
| Joplin_24_N | 35 | 41 | 24 | Grassland | 2010-2020 | 37.43 | -94.58 |
| Buffalo_13_ESE | 31 | 44 | 25 | Grassland | 2012-2023 | 45.52 | -103.30 |
| ClotdelesPeresII | 19 | 49 | 32 | Cropland | 2021-2023 | 42.16 | 0.84 |

[Figure]

Figure R1. The geographical locations of thirty sites.

Table R2. The average $R^2$ values between the predictions (1, 3, and 7 days) generated by the evaluated models and the ground truth across thirty sites at five depths.

| Depth/m | LSTM | | | CNN | | | Transformer | | |
|---|---|---|---|---|---|---|---|---|---|
| | 1d | 3d | 7d | 1d | 3d | 7d | 1d | 3d | 7d |
| 0.05 | **0.943** | **0.895** | **0.816** | 0.939 | 0.884 | 0.793 | 0.933 | 0.886 | 0.805 |
| 0.10 | 0.954 | **0.909** | 0.838 | **0.956** | 0.909 | 0.826 | 0.949 | 0.906 | **0.839** |
| 0.20 | **0.963** | **0.916** | 0.842 | 0.961 | 0.912 | 0.823 | 0.952 | 0.912 | **0.843** |
| 0.50 | **0.937** | **0.873** | **0.749** | 0.909 | 0.702 | 0.532 | 0.917 | 0.840 | 0.716 |
| 1.00 | **0.944** | 0.878 | 0.746 | 0.919 | 0.811 | 0.547 | 0.939 | **0.879** | **0.758** |
| | LSTM-CNN | | | CNN-LSTM | | | CNN-with-LSTM | | |
| | 1d | 3d | 7d | 1d | 3d | 7d | 1d | 3d | 7d |
| 0.05 | **0.939** | **0.889** | **0.809** | 0.936 | 0.885 | 0.800 | 0.936 | 0.880 | 0.792 |
| 0.10 | 0.950 | **0.901** | 0.820 | 0.943 | 0.895 | **0.821** | **0.951** | 0.899 | 0.810 |
| 0.20 | **0.959** | **0.906** | **0.822** | 0.952 | 0.899 | 0.816 | 0.950 | 0.891 | 0.795 |
| 0.50 | **0.916** | **0.814** | **0.683** | 0.867 | 0.715 | 0.546 | 0.886 | 0.782 | 0.644 |
| 1.00 | 0.908 | 0.788 | 0.546 | **0.908** | **0.821** | **0.651** | 0.897 | 0.787 | 0.575 |
| | FA-LSTM | | | TA-LSTM | | | FTA-LSTM | | |
| | 1d | 3d | 7d | 1d | 3d | 7d | 1d | 3d | 7d |
| 0.05 | **0.944** | **0.902** | **0.827** | 0.937 | 0.888 | 0.809 | 0.942 | 0.897 | 0.823 |
| 0.10 | **0.960** | **0.921** | **0.848** | 0.950 | 0.899 | 0.826 | 0.950 | 0.906 | 0.839 |
| 0.20 | **0.965** | **0.925** | **0.849** | 0.957 | 0.909 | 0.823 | 0.954 | 0.907 | 0.825 |
| 0.50 | **0.949** | **0.881** | **0.770** | 0.923 | 0.869 | 0.745 | 0.870 | 0.773 | 0.653 |
| 1.00 | **0.947** | **0.896** | **0.794** | 0.927 | 0.854 | 0.703 | 0.915 | 0.842 | 0.672 |
| | LSTM | | | GAN-LSTM | | | | | |
| | 1d | 3d | 7d | 1d | 3d | 7d | | | |
| 0.05 | 0.943 | 0.895 | 0.816 | **0.944** | **0.897** | **0.819** | | | |
| 0.10 | 0.954 | 0.909 | **0.838** | **0.956** | **0.910** | 0.838 | | | |
| 0.20 | 0.963 | 0.916 | 0.842 | **0.963** | **0.919** | **0.846** | | | |
| 0.50 | 0.937 | 0.873 | 0.749 | **0.946** | **0.893** | **0.777** | | | |
| 1.00 | 0.944 | 0.878 | 0.746 | **0.948** | **0.896** | **0.793** | | | |

From the results, we could also draw similar conclusions: LSTM and Transformer (Encoder) demonstrate greater stability when making long-term or deep soil moisture predictions, while CNNs offer advantages in computational costs. In the case of hybrids combining CNN and LSTM, LSTM-CNN slightly outperforming the CNN-LSTM and CNN-with-LSTM. However, the benefits of this combination approach are not significant for soil moisture prediction. Regarding introducing attention mechanisms to enhance predictions, it is evident that the FA-LSTM still remains the superior performance. Furthermore, when comparing the LSTM with GAN-LSTM, adversarial training strategies indeed improve the performance of the LSTM in most scenarios. We have incorporated the results of these new calculations into the revised manuscript.

3. Regarding the model's transferability, how does it perform when trained in one region and predicted in another? Research limited to a small area might restrict its applicability. It's suggested

that the authors use more sites for further validation.

Response:

Thank you for your comments. Our primary objective is to understand the data utilization of various deep learning models emphasizing the focus on soil moisture dynamics and spatio-temporal patterns rather than purely mathematical perspectives (Jiang and Li, 2023; Wang et al., 2023). Therefore, we trained the model using the data from one site and examined the model at the same site. Transferability is not investigated in this study. We have added an explanation in the manuscript to clarify this.

We agree that the transferability of models is very important and warrants further research, and our work can serve as the basis for future research. In the future, we should try to input static properties and make suitable designs in model structures, enriching the incomplete descriptions by static properties with a small amount of in-situ observations and enhancing the model transferability.

4. It is recommended that the authors establish a benchmark to show the model's performance and compare it to the results of other peers.

Response:

Thank you for your suggestions. We acknowledge the importance of benchmark owing to the variations in input data and settings in soil moisture prediction research. We have conducted comparisons for several classical models, such as the RF (Carranza et al., 2021), ELM (Liu et al., 2014), LSTM (Fang et al., 2019), attention mechanisms with LSTM (Li et al., 2022a) and CNN-with-LSTM (Yu et al., 2021). These models are widely recognized by peers in the research community for their suitability in processing time-series data. In our work, we also firstly proposed some new models for soil moisture prediction, such as the Transformer model and the GAN-LSTM. One goal of our research is to provide a comprehensive benchmark for soil moisture prediction with deep learning using in-situ observations.

5. Can the authors provide the training and testing time of the model?

Response:

Thank you for your suggestions. The training time for every 100 epochs and the required training epoch numbers of LSTM, CNN, Transformer, and their hybrid variants are recorded in Table R3. With the same training epoch numbers, CNN-with-LSTM exhibits the longest total training time at 166.56 seconds, while the Transformer shows the shortest training time at 73.11 seconds. The testing time for each model is within a few seconds.

Table R3. The training time for all the research deep learning models.

| model | | LSTM | | CNN | | Transformer | | CNN-LSTM | | LSTM-CNN | |
|---|---|---|---|---|---|---|---|---|---|---|---|
| Time/100epoch | Time | 8.93 | 133.98 | 5.93 | 88.89 | 4.87 | **73.11** | 9.25 | 138.72 | 10.56 | 158.40 |
| Epoch num | total | 1500 | | 1500 | | 1500 | | 1500 | | 1500 | |
| model | | CNN-w-LSTM | | FA-LSTM | | TA-LSTM | | FTA-LSTM | | LSTM-GAN | |
| Time/100epoch | Time | 11.10 | **166.56** | 9.37 | 140.52 | 8.69 | 130.29 | 9.84 | 147.57 | 9.28 | 139.14 |
| Epoch num | total | 1500 | | 1500 | | 1500 | | 1500 | | 1500 | |

6. There is a problem with the meteorological data reference. For the meteorological data, the authors still need to specify the data source. The resolution for some variables seems coarse. Authors

can use the following documentation to get more details:

https://power.larc.nasa.gov/docs/methodology/data/sources/

Response:

Thank you for your suggestions. We have provided the following detailed information about the data source and emphasize the limitations of the variable resolution in the manuscript.

Data Source: Specifically, the meteorological data applied in this work is sourced from the NASA POWER project (https://power.larc.nasa.gov/), which provides a wide range of meteorological data, including temperature, precipitation, solar radiation, and more.

Data Resolution Issue: The horizontal resolution of the primary solar data source (longwave and shortwave downward irradiance) is a global 1° x 1° latitude/longitude grid while the meteorological data sources (air temperature, relative humidity, wind speed, and precipitation) are ½° x ⅝° latitude/longitude grid. Detailed information can be found at (https://power.larc.nasa.gov/data-access-viewer/). The temporal resolutions of the data sets are daily. The meteorological data are used as an auxiliary component for soil moisture prediction in our work. Therefore, even though the resolution of some variables appears coarse, we can safely disregard the potential influence of resolution on our research findings and conclusions.

7. Relying solely on time series data may not be sufficient. Could other attributes be introduced, such as vegetation and soil information? Such additions might help improve the model's performance.

Response:

Thank you for your suggestions. Static attributes such as vegetation land cover and soil information are indeed necessary for generating transferable model predictions. There are several reasons why we do not consider static properties in our work: 1) Given our research objectives, which aim to understand how different deep learning model structures adapt to spatio-temporal variations in soil moisture data, introducing coarse static attributes will make the question more complicated. The static characteristics of different sites are finally captured by the model within the network parameters through training;

2) Static attributes derived from site information cannot completely describe the local situation, such as root water uptake, preferential flow, etc. An ideal transferable model should encompass both static properties and the incorporation of limited local in-situ data. This represents a highly meaningful area of research that can serve as a primary focus for future studies.

8. For the Transformer model, with time series data, relying solely on positional embeddings might make it difficult to reflect seasonality and long-term sequential information. How did the authors address this issue?

Response:

Thank you for your comments. Because of the nonlocal effect in the Transformer, appropriate positional encoding is essential (Vaswani et al., 2017; Devlin et al., 2019; Shaw et al., 2018). Positional encodings enable the encoding of absolute or relative information in sequential inputs as required. Our work considers that the soil moisture on the fifth day can be entirely determined by the conditions of the first four days, which is reasonable from the perspective of soil moisture dynamics. In this case, the provided absolute positional encoding is adequate.

When addressing seasonality, absolute encoding, such as the sinusoidal positional encodings, is

generally acknowledged to have the ability to encode this kind of long-term sequential information (Devlin et al., 2019). Additionally, the incorporation of time-dependent variables into the model can further enhance its capacity to capture seasonality and long-term dependencies.

9. The authors conducted a SHAP analysis. Under what conditions does the model perform well? Can feature analysis determine which features enhance the model's performance? How does it differ from the feature importance of Random Forest? Do both have similar conclusions?

Response:

Thank you for your comments. Incorporating the model performances and SHAP analysis results, we believe that the SHAP analysis of a well-performing model should be as follows:

1) Strongly reflecting on important characteristics, such as LSTM and CNN, leads to better results in surface soil moisture or short-term prediction.

2) Avoiding overlearning unimportant features ensures that established correlations remain sufficiently accurate. False correlations tend to complement each other, leading to the learning of incorrect rules and resulting in poor performance for long-term forecasts.

Because input features have varying effects on soil water prediction results across different scenarios (sites, depths), there is no standardized format for SHAP analysis. However, the SHAP analysis can help us discover the characteristics of various basic and hybrid neural networks in processing soil moisture data.

Generally, the results of feature importance ranking based on SHAP analysis are consistent with that of feature importance analysis of Random Forest. SHAP calculates the feature importance rooted in the concept of Shapley value (in Appendix B), while Random Forest determines feature importance by evaluating the reduction in impurity when each feature is split in the tree. Both of them can reflect the feature importance.

However, sometimes the feature importance ranking results may be different. SHAP provides a more detailed and comprehensive calculation of the influence of each feature on the prediction result, with high computational costs. In contrast, Random Forest's feature importance estimation is a more general measure. In addition, SHAP technology can also detect the interaction between features, thus providing a more comprehensive and refined feature importance ranking result, as depicted in Figure R2. Each point shows the Shapley value (impact on the results) of a specific feature in a sample, with the color indicating the value of the input feature. In Figure R3, the feature importance rankings perform consistently on significant features but exhibit slight differences on less significant factors, indicating that both feature analysis methods are credible.

The selection of the two methods depends on the research requirements. If the emphasis is on the relative importance of input features, feature importance from Random Forest is sufficient. However, if a detailed explanation of individual forecasts is required, SHAP analysis may be more suitable. Generally, a combined utilization of the two methods can be employed to achieve a more comprehensive model understanding. In our work, SHAP analysis is more suitable because of the interaction between the input features and the need for detailed insights.

We have made clearer expressions about the SHAP analysis in the revised manuscript.

[Figure]

Figure R2. SHAP value analysis for Random Forest at the Monahans site at 0.0500m

[Figure]

Figure R3. Mean |SHAP value| bar analysis (a) and feature importance analysis (b) for Random Forest at the Monahans site at 0.0500m

10. (~Line 595) In Figure 13, what do the x and y axes represent? Does the input data consist solely of soil moisture or does it also include other time series data? What does the distribution shape of points in the model signify? Can the authors infer model performance or the most critical variables from the results? How do the authors explain the phenomena in the figure? What caused these phenomena?

Response:

Thank you for your comments. We have added more details about the t-SNE visualizations in the manuscript.

1) When conducting t-SNE visualizations, the x and y axes make no sense, as depicted in Figure R4 of Van der Maaten and Hinton's work (2008). Only the relative distance between sample points matters.

2) The input data in Figure 13(a) from the manuscript denotes the flattened form of the four days' inputs $I, \{x_{t-3}, x_{t-2}, x_{t-1}, x_t\}$, where $x_t = \{P_t, T_t, LW_t, SW_t, RH_t, WS_t, ST_t, SM_{t-1}\}$, mentioned in Line 128.

3) The closer the points are to each other, the higher the similarity is. The color of each sample point corresponds to the soil moisture content value (Line 605). Based on this, the distribution of points should be analyzed from two aspects: the regularity of color distribution from light to dark; and the specific shape generated through models. The regularity of the colored plots exhibits two forms: linearly separable or curved. We infer that the linearly separable pattern holds greater utility in this soil moisture regression task. The specific shape reveals the distinct encoding strategies of models, and it appears that the points by more complex models tend to cluster. However, it appears that this clustering is not the primary determinant of accuracy in regression tasks. We could observe that

models with specific shapes in t-SNE visualization do not perform the superior accuracy, such as Figure 13(e) (g) (j).

4) For a soil moisture prediction regression task, we infer that in t-SNE visualizations of models with great forecasting capacity, the sample points can be arranged vertically from light to dark in color, such as the Figure 13(h). Additionally, these visualizations enable us to discern the impact of the attention mechanism and adversarial training on LSTM in Figure 13(k)(h), ultimately leading to enhanced accuracy.

5) Generally, from the t-SNE visualizations, it can be summarized that different deep learning models capture distinct intrinsic characteristics of input data and encode them into various vectors for making predictions. We analyze the models from the perspective of soil moisture prediction tasks, and we believe that this phenomenon is likely associated with the complexity and data utilization characteristics. Further validation through mathematical research will be necessary to substantiate this hypothesis in the future.

[Figure]

(a) Visualization by t-SNE.

(b) Visualization by Sammon mapping.

Figure 2: Visualizations of 6,000 handwritten digits from the MNIST data set.

Figure R4 The t-SNE Visualizations of 6,000 handwritten digits from the MNIST data set from Van der Maaten and Hinton (2008)

11. (~Line 305) "To enhance the accuracy of deep learning models and address the issue of lack of interpretability, attention mechanisms have been incorporated into LSTM models to weigh the importance of different." Can the authors specify how the Attention mechanism addresses this issue?
Response:

Thank you for your comments. When applying an attention mechanism into the LSTM model at feature (FA-LSTM) or time dimensions (TA-LSTM), it will generate attention weights for the input features $\{\alpha1_t, \alpha2_t, ..., \alpha n_t\}$ or the temporal hidden states $\{\beta_1, \beta_2, \beta_3, \beta_4\}$ through training, as illustrated in Figure R5. These attention weights enable the model to assign importance to various elements within the input sequence, allowing it to select the most crucial factors for making more accurate predictions.

Additionally, these attention weights offer a visualized representation, as demonstrated in Figure 10 of the manuscript, which provides insights into the sections of the input sequence most essential for a specific prediction. The feature importance depicted in Figure 10 represents the attention weights

generated through the self-learning of the attention mechanism. Higher feature importance values signify that the corresponding features are more significant within the model. Moreover, in our work, the feature importance in Figure 10 shows a reasonable adaptation to the varying depth, demonstrating the effective feature selection capability of attention mechanisms (Line 515). This visualization facilitates a deeper understanding of how the model learns and utilizes data, contributing to a more comprehensive comprehension of the deep learning model's decision-making process.

[Figure]

Figure R5 Structures of feature attention mechanisms and temporal attention mechanisms.

**Code problems:**

1. There is an issue with the code. It can not run directly and requires reader debugging.

We feel very sorry for the code problems. We have reorganized and uploaded the codes for readers to use. (https://doi.org/10.5281/zenodo.10060492)

2. Why did the authors choose only about five years of data for training and testing?

Thank you for your comments. The duration for training and testing the model is determined based on the available data from ISMN. When collecting data, the data quality is our focus, so the length of the dataset may vary for each site. As we mentioned in Line 388 in the manuscript, the collected data in Section 2 is divided into training, validation, and test sets in a 6:2:2 ratio in time order. This allocation process is implemented in our code at Lines 71-72.

3. It seems that the code didn't utilize GPU. With only a CPU, every 100 epochs take about 20 seconds. With GPU acceleration, computation cost comparison holds more significance.

Thank you for your suggestions. The utilization of GPUs indeed improves the training speed. We have clarified in the revised manuscript that the computational costs are evaluated using CPUs. In the newly uploaded codes, we have provided a GPU version. Besides, we conducted a comparison of the training speed between CPU and GPU, as illustrated in Figure R6. It is evident that almost all models benefit from significantly improved training speed when utilizing a GPU. Notably, the transformer model in our study also exhibits fast training performance when running on a CPU.

[Figure]

Figure R6 Comparisons of training time for every 100 epochs of ten models with CPU and GPU.

4. During testing, the authors set the batch_size to 1, which is not efficient for large datasets.

Thank you for your comments. As previously stated in Line 388, the collected data is split into training, validation, and test sets following a 6:2:2 ratio in time order. For each result, all the data in the test set (20% of the collected data) are evaluated. The test batch_num is 1, but the batch_size encompasses all the available data in the test set.

5. The code seems to use four days of data to forecast the next day, but the paper needs to explicitly mention whether the authors use 4 days of historical data to forecast the next 1, 3, and 7 days.

Thank you for your suggestions. In the code, we conducted soil moisture predictions iteratively. After completing the prediction for the first day, we utilize the generated soil moisture data for the first day, along with the corresponding observed meteorological data on that day, combined with historical three-day data. This reconstructed the new four-day input, which is used to predict soil water for the second day, and this process is repeated to complete predictions for 1, 3, and 7 days, as stated in Line 382. We have made clearer descriptions in the revised manuscript.

6. Each forecasting step overlaps when forecasting for 3 or 7 days. How did the authors handle these overlapping data to calculate metrics?

Thank you for your comments. When calculating metrics for 3 or 7 days, we compute them by comparing all the predictions with the ground truth. This is implemented in our code at Lines 223-225.

7. After normalizing the data, did the authors denormalize it when calculating metrics?

Thank you for your comments. When calculating metrics, it is necessary to denormalize the data. In our code, this denormalization is performed at the final stage at Lines 215-216. It's worth noting that the unit for soil moisture, used in the calculation of metrics, is expressed in percentage (%), according to the work of Gill et al. (2006).

**Reference:**

Abbaszadeh, P., Moradkhani, H., and Zhan, X.: Downscaling SMAP radiometer soil moisture over the CONUS using an ensemble learning method, Water Resour. Res., 55, 324–344, 2019.

Carranza, C., Nolet, C., Pezij, M., and van der Ploeg, M.: Root zone soil moisture estimation with Random Forest, J. Hydrol., 593, 125840, https://doi.org/10.1016/j.jhydrol.2020.125840, 2021.

Datta, P. and Faroughi, S. A.: A multihead LSTM technique for prognostic prediction of soil moisture, Geoderma, 433, 116452, https://doi.org/10.1016/j.geoderma.2023.116452, 2023.

Devlin, J., Chang, M. W., Lee, K., and Toutanova, K.: BERT: Pre-training of deep bidirectional transformers for language understanding, NAACL HLT 2019 - 2019 Conf. North Am. Chapter Assoc. Comput. Linguist. Hum. Lang. Technol. - Proc. Conf., 1, 4171–4186, 2019.

Fang, K., Pan, M., and Shen, C.: The Value of SMAP for Long-Term Soil Moisture Estimation with the Help of Deep Learning, IEEE Trans. Geosci. Remote Sens., 57, 2221–2233, https://doi.org/10.1109/TGRS.2018.2872131, 2019.

Feng, D., Beck, H., de Bruijn, J., Sahu, R. K., Satoh, Y., Wada, Y., Liu, J., Pan, M., Lawson, K., and Shen, C.: Deep Dive into Global Hydrologic Simulations: Harnessing the Power of Deep Learning and Physics-informed Differentiable Models (δHBV-globe1. 0-hydroDL), Geosci. Model Dev. Discuss., 2023, 1–23, 2023.

Gill, M. K., Asefa, T., Kemblowski, M. W., and McKee, M.: Soil moisture prediction using support vector machines, J. Am. Water Resour. Assoc., 42, 1033–1046, https://doi.org/10.1111/j.1752-1688.2006.tb04512.x, 2006.

Jiang, H. and Li, Q.: Approximation theory of transformer networks for sequence modeling, arXiv Prepr. arXiv2305.18475, 2023.

Li, Q., Hao, H., Zhao, Y., Geng, Q., Liu, G., Zhang, Y., and Yu, F.: GANs-LSTM Model for Soil Temperature Estimation from Meteorological: A New Approach, IEEE Access, 8, 59427–59443, https://doi.org/10.1109/ACCESS.2020.2982996, 2020.

Li, Q., Zhu, Y., Shangguan, W., Wang, X., Li, L., and Yu, F.: An attention-aware LSTM model for soil moisture and soil temperature prediction, Geoderma, 409, 1–17, https://doi.org/10.1016/j.geoderma.2021.115651, 2022a.

Li, Q., Li, Z., Shangguan, W., Wang, X., Li, L., and Yu, F.: Improving soil moisture prediction using a novel encoder-decoder model with residual learning, Comput. Electron. Agric., 195, https://doi.org/10.1016/j.compag.2022.106816, 2022b.

Liu, J., Rahmani, F., Lawson, K., and Shen, C.: A multiscale deep learning model for soil moisture integrating satellite and in situ data, Geophys. Res. Lett., 49, e2021GL096847, 2022.

Liu, Y., Mei, L., and Ki, S. O.: Prediction of soil moisture based on Extreme Learning Machine for an apple orchard, CCIS 2014 - Proc. 2014 IEEE 3rd Int. Conf. Cloud Comput. Intell. Syst., 400–404, https://doi.org/10.1109/CCIS.2014.7175768, 2014.

Van der Maaten, L. and Hinton, G.: Visualizing data using t-SNE., J. Mach. Learn. Res., 9, 2008.

Orth, R.: Global soil moisture data derived through machine learning trained with in-situ measurements, Sci. Data, 8, 1–14, 2021.

Orth, R., Weber, U., and Park, S. K.: High-resolution European daily soil moisture derived with machine learning (2003–2020), Sci. Data, 9, 1–13, 2022.

Shaw, P., Uszkoreit, J., and Vaswani, A.: Self-attention with relative position representations, NAACL HLT 2018 - 2018 Conf. North Am. Chapter Assoc. Comput. Linguist. Hum. Lang. Technol. - Proc. Conf., 2, 464–468, https://doi.org/10.18653/v1/n18-2074, 2018.

Vaswani, A., Shazeer, N., Parmar, N., Uszkoreit, J., Jones, L., Gomez, A. N., Kaiser, Ł., and Polosukhin, I.: Attention is all you need, Adv. Neural Inf. Process. Syst., 30, 2017.

Wang, S., Li, Z., and Li, Q.: Inverse Approximation Theory for Nonlinear Recurrent Neural Networks, arXiv Prepr. arXiv2305.19190, 2023.

Yu, J., Zhang, X., Xu, L., Dong, J., and Zhangzhong, L.: A hybrid CNN-GRU model for predicting soil moisture in maize root zone, Agric. Water Manag., 245, 106649, https://doi.org/10.1016/j.agwat.2020.106649, 2021.

---

## Author Response (AR2)

Dear Editor Fadji Zaouna Maina:

    We are grateful to receive comments and suggestions on our manuscript of 'HESS-2023-177' from you and the reviewers. We have carefully considered these comments and made revisions. Furthermore, we have adjusted the color schemes for Figures 1 and 3. All the modifications in the revised manuscript are highlighted in blue color. The responses to the reviewer's comments are also appended for your convenience.

We hope the revision will be satisfactory. If you have any further questions or concerns, please let us know. Once again, we appreciate your time on this manuscript.

Yours sincerely,

Liangsheng Shi

Wuhan University

Thank you for your response and the revisions undertaken. Regarding the content of your response, I have some minor questions and suggestions that require further clarification and consideration:

Response:

We truly appreciate the time and effort you invested in providing constructive feedback. Your valuable insights have significantly contributed to the improvement of our paper. We have considered the suggestions and carefully reworded expressions to improve the manuscript.

1. Concerning the question I previously raised: "Can the authors provide the training and testing times of the model?" I intended to inquire about the specific dates used for the training and testing datasets.

Response 1:

Thank you for your comment. The specific dates used for the training, validation, and testing datasets at each site are summarized in Table D2 in the Appendix D.

2. Suggestions: The static attribute data are not only important for the model's transferability but also crucial for the model to capture the spatiotemporal variation in soil moisture. Static attributes can provide the model with additional dimensions of information. For example, under identical precipitation conditions, the soil moisture content typically differs between sandy and clayey soils. Without including this additional information, or not using a larger model, even a trained model might struggle to accurately capture the static features of different sites.

Response 2:

Thank you for your comment. The static attribute holds significant importance in capturing spatiotemporal variation in soil moisture within the context of physical modeling. However, in data-driven models that neglect transferability, such as a model specifically trained for a particular site, providing static attributes as inputs may show no apparent benefits for the model.

3. Since the authors mentioned using data from only the past four days, is the term "long-term prediction" used in the article accurately precise? A week's duration is difficult to be referred to as a long time series prediction.

Response 3:

Thank you for your comment. We acknowledge that the term "long-term prediction" used in Line 474, 479, 576, 681, and 688 is not appropriate. Consequently, we have replaced it with either "7-day prediction" or "weekly prediction."

4. Although positional encodings are an indispensable part of the Transformer model, their capability to handle long-term dependencies might require additional technical support or modifications to the standard Transformer model.

Response 4:

Thank you for your comment. We agree that additional technical modifications may be necessary for the standard Transformer model when applied to long-term tasks. The core component of the Transformer, self-attention, holds potential for modeling long-term dependencies, as demonstrated in the study by Wang et al. (2018). We hypothesize that incorporating time-dependent variables or devising specific positional encodings and model structures could enhance the model's capacity in

capturing seasonality and long-term dependencies.

**Reference:**

Wang, X., Girshick, R., Gupta, A., and He, K.: Non-local neural networks, in: Proceedings of the IEEE conference on computer vision and pattern recognition, 7794–7803, 2018.